# Future changes in regional inter-monthly precipitation patterns of the East Asian summer monsoon and associated uncertainty factors

Yeon-Hee Kim[1], Seung-Ki Min[1,2]

[1]Division of Environmental Science and Engineering, Pohang University of Science and Technology, Pohang, 37673, South Korea
[2]Institute for Convergence Research and Education in Advanced Technology, Yonsei University, Incheon, 21983, South Korea

*Correspondence to*: Seung-Ki Min (skmin@postech.ac.kr)

**Abstract.** East Asia has been identified as a key area at risk of precipitation increases resulting from global warming. East Asian summer monsoon has distinct regional inter-monthly precipitation patterns and the simulation characteristics of global climate models therefore needs to be evaluated closely to obtain a reliable projection of future precipitation patterns and associated extreme events. Using metrics for the inter-monthly variability of monsoon precipitation over East Asia, this study evaluated the performance of Coupled Model Intercomparison Project Phase 6 (CMIP6) models and analyzed future projections and uncertainty factors. Regional inter-monthly precipitation patterns were simulated reasonably well by CMIP6 models but with weaker rainfall amplitudes. CMIP6 models simulated more intense precipitation than the predecessor CMIP5 ones and captured observations better. In future projections, an overall precipitation increase occurs during both the northward migration of the rain band and during peak monsoon time over East Asia and the three subregions, with stronger changes under higher emission scenarios. This increase was mainly ascribed to a thermodynamic response due to increased moisture availability in line with global warming. Internal climate variability and model uncertainty dominated future precipitation uncertainties, which are associated with tropical ocean warming patterns. Dynamic terms explained a large portion of model uncertainty due to circulation changes, and thermodynamic terms were significantly related to scenario uncertainty.

## 1 Introduction

The East Asian summer monsoon is an important large-scale circulation system occurring across countries such as China, Korea, and Japan. The monsoonal rain band moves northward over East Asia from May to July, and its precipitation exhibits complex spatiotemporal patterns influenced by internal climatic variability (e.g., influences of the El Niño–Southern Oscillation) and topography (e.g., the Tibetan Plateau). The Intergovernmental Panel on Climate Change (IPCC) identified East Asia as one of the key risk regions where statistically significant climate change causing heavy precipitation can be expected as a result of a 1.5 °C or, even greater consequence, a 2 °C global warming scenario (Hoegh-Guldberg et al., 2018). Such change would deliver a huge socioeconomic impact on the densely populated region. Consequently, to establish a reliable projection of future shifts in precipitation over East Asia, the performance of global climate models (GCMs) must be evaluated closely, and further climatic projections alongside uncertainty assessments are essential.

A new generation of GCMs has been released for the Coupled Model Intercomparison Project Phase 6 (CMIP6) experiments (Eyring et al., 2016). The CMIP6 ensembles comprise a complex range of models, from GCMs to Earth System Models, with improved physical parameterization and finer spatial resolution than earlier models (Eyring et al., 2016; Marotzke et al., 2017). Recent studies have confirmed that the CMIP6 models provide an advanced simulation of monsoon precipitation compared to CMIP5 ones, especially in terms of precipitation intensity (Chen et al., 2021; Jiang et al., 2020; Kim et al., 2020; Wang et al., 2021; Xin et al., 2020). CMIP6 models can simulate a more realistic climatological pattern of the East Asian summer monsoon, better capturing the northward movement of the monsoonal rain band due to finer horizontal resolutions (Chen et al., 2021; Jiang et al., 2020) and smaller model biases in sea surface temperature (SST) over the Northwestern Pacific Ocean (Xin et al., 2020).

The Asian summer monsoon domain is one of the region most vulnerable to global warming (IPCC, 2013). An increased mean and extreme precipitation have been consistently projected for this region under Representative Concentration Pathway (RCP) scenarios (IPCC, 2013; Kitoh et al., 2013; Endo and Kitoh 2014; Freychet et al., 2015; Li et al., 2015; Lee et al., 2017; Park and Min 2019). Recent studies based on the CMIP6 ensemble have also predicted that precipitation will increase over East Asian monsoon regions by about 8–14% under four Shared Socioeconomic Pathway (SSP) scenarios (SSP1-2.6, SSP2-4.5, SSP3-7.0, and SSP5-8.5) and by 5.6% $°C^{-1}$ across a long-term period (2081–2100) under the SSP5-8.5 scenario (Chen et al., 2020; Ha et al., 2020; Moon and Ha 2020. They also suggested that the thermodynamic effects resulting from increased moisture levels may play a dominant role in elevating precipitation levels while the dynamic effects from circulation changes contribute more to the uncertainty of future precipitation projection (Zhou et al., 2020). These results are consistent with those produced via CMIP5 models  (Endo and Kitoh 2014; Lee et al., 2017; Zhou et al., 2019a, 2019b).

Although model simulation performance has improved, uncertainties remain regarding future precipitation projections. These uncertainties tend to increase at a smaller regional scale and with stronger future radiative forcing (Chen et al., 2020). Sources of uncertainty in climate projections consist of three factors: anthropogenic forcing agents, parametric and structural model uncertainties related to the response of the climate model to specified forcing agents, and natural internal variability intrinsic to the climate system (Hawkins and Sutton 2009, 2011). In global mean precipitation projections using the CMIP6 ensemble, model uncertainty plays the dominant role of these three factors, with internal variability and scenario uncertainty contributing to the total uncertainty in the near term (2021–2040) and long term (2081–2100), respectively (Lehner et al., 2020). For the global land monsoon area, model uncertainty dominates the projection uncertainty (~90%), whereas the contribution of internal variability decreases and that of scenario uncertainty increases over time via the CMIP5 ensemble (Zhou et al., 2020).

Previous CMIP6 analyses of monsoon precipitation have focused on summer mean precipitation within the monsoon region. Understanding the changes in regional inter-monthly precipitation patterns along the monsoon band is important. However, comprehensive research is lacking on model performance, future projections, and uncertainty in this regard. The objectives of the present study are as follows: 1) to compare the performance of CMIP6 and CMIP5 models in simulating inter-monthly evolution of precipitation patterns in East Asian subregions; 2) to investigate projections of regional inter-

monthly precipitation patterns and sources of model uncertainty using CMIP6 models under four SSP scenarios; and 3) to examine the uncertainty factors and associated physical mechanisms of East Asian subregional precipitation changes.

## 2 Data and methods

### 2.1 Data

We used daily precipitation data from 25 CMIP6 (Table S1; Eyring et al., 2016) and 22 CMIP5 models (Table S2; Taylor et al., 2012). These models were selected based on their availability of data regarding daily precipitation. To match future forcing levels, we selected the SSP1-2.6, SSP2-4.5, and SSP5-8.5 scenarios from CMIP6 (O'Neil et al., 2016) and the RCP2.5, RCP4.5, and RCP 8.5 scenarios from CMIP5 (Taylor et al., 2012). For model evaluation, historical simulations were used for CMIP6, and historical simulations for CMIP5 were extended to 2014 by combining them with the RCP 4.5 simulations. The evaluation and reference period was selected as 1995–2014. We used the first realization for each model and each scenario, and we present projections for three specific future periods, namely, the near term (NT; 2021–2040), mid term (MT; 2041–2060), and long term (LT; 2081–2100).

The CMIP5 and CMIP6 historical simulations were evaluated against pentad precipitation data retrieved from the Global Precipitation Climatology Project (GPCP; Adler et al., 2018) and Climate Prediction Center Merged Analysis of Precipitation (CMAP; Xie and Arkin 1997) datasets. All observations and model precipitation data were re-gridded into a 2.5° × 2.5° resolution prior to further analysis

### 2.2 Indices of regional inter-monthly precipitation patterns

To analyze rainfall pattern changes across East Asia, three East Asian subregions (Fig. S1) were defined as China (20–45°N, 110–120°E), Korea (20–45°N, 120–130°E), and Japan (20–45°N, 130–142°E), following Kusunoki and Arakawa (2015). The whole East Asian domain encompasses the domain of 15–50°N, 100–150°E. To define precipitation indices that can explain intra-seasonal evolution patterns, the rainy season was classified into two indices according to the movement of the monsoonal rain band. Figure 1 shows the latitude–time cross-section of precipitation climatology that depicts the zonal averages for East Asia and the three subregions, as derived from GPCP, CMAP, CMIP6 Multi-Model Ensemble (MME), and CMIP5 MME data. In general, from mid-May to June, the monsoon band moved northward from lower latitudes (around 20–25°N), reaching the mid-latitudes (around 30–35°N) by July (Fig. 1).

We defined our two precipitation indices by averaging the precipitation at (1) the time of northward movement of the monsoon band and (2) the peak of the monsoon band. An index of northward migration (hereafter referred to as "northward migration") was defined as the average precipitation during June between 20°N and 35°N in the entire East Asian region, 20°N and 32.5°N in China, and 25°N and 35°N in both Korea and Japan (red boxes in Fig. 1). The peak time index (hereafter referred to as "peak time") was defined as the average precipitation during July between 30°N and 40°N in East Asia, 27.5°N and

37.5°N in China, and 32.5°N and 42.5°N in both Korea and Japan (blue boxes in Fig. 1).To examine the meridional movement of the monsoon band under different climate change scenarios, we defined the monsoon band location. The fourth-order centered difference was used to calculate the meridional precipitation gradient, and the latitude with maximum precipitation ($d$ precipitation / $d$ latitude = 0) was determined using linear interpolation.

The developed indices were first evaluated in relation to East Asia precipitation patterns (Fig. S2a,b). Figure S1 shows the regression pattern of the northward migration and peak time index over East Asia and precipitation using GPCP data from 1995 to 2014. The regression patterns reveal the movement of the monsoon precipitation band during the northward migration and peak time, indicating that these indices are suitable for representing the inter-monthly evolution of monsoon rain band in East Asia.

For further evaluations of the indices, we examined the relationships between two precipitation-based indices and two East Asia summer monsoon indices: East Asia summer monsoon index [EASMI; defined as the difference between the 850hPa zonal wind anomalies averaged over the southern (100-150°E, 10-20°N) and northern (100-150°E, 25-35°N) regions; Zhang et al., 2003] and western North Pacific subtropical high [WNPSH; defined as the 850hPa eddy geopotential height averaged over 120°-150°E, 15-30°N; Zhou et al., 2020]. The EASMI shows a statistically significant negative correlation with the peak time index over East Asia (r=-0.49 for GPCP, r=-0.46 for GMAP) and with the northward migration index over China
(r=-0.45 for GPCP). In contrast, the WNPSH exhibits a strong positive correlation with the northward migration index over China (r=0.63 for GPCP, r=0.50 for CMAP). However, their correlations with indices for Korea and Japan are generally weak and not statistically significant, suggesting that these circulation-based indices have limited ability to capture regional monsoon characteristics.

This is because these summer monsoon indices are based on the large-scale atmospheric circulation during the East Asia summer, and therefore have limitations in explaining regional rainfall mechanisms and intra-seasonal variability. Figure S2c and d show the regression patterns of the northward migration and peak time indices over East Asia with the 850hPa zonal wind, which is used to calculate the EASMI. During the peak time over East Asia, zonal wind anomalies in the two regions exhibit a strong correlation with the index, whereas no significant correlation is observed during the northward migration.
Figure S2e and f show the regression patterns of the northward migration and peak time indices over China with the 850hPa eddy geopotential height, which is used to calculate the WNPSH. The 850hPa eddy geopotential height shows a strong correlation during the northward migration over China but a weaker correlation during the peak time. Overall, our proposed indices for inter-monthly precipitation evolutions have the advantage of directly reflecting precipitation changes, better representing regional features, and allowing for quantitative analysis of the intra-seasonal evolution of monsoon rain band over
East Asia.

**2.3 Uncertainty partitioning**

To examine the source of uncertainty in future precipitation projection, we employed the method developed by Hawkins and Sutton (2009, 2011) where total uncertainty ($T$) consists of the internal variability ($I$), model uncertainty ($M$), and scenario uncertainty ($S$). Each term can be estimated as a variance across a given time ($t$) as follows:

$$T(t) = I(t) + M(t) + S(t) \tag{1}$$

This equation assumes that sources of uncertainty are independent. The fractional uncertainties (%) were furthermore calculated as $I(t)/T(t)$, $M(t)/T(t)$, and $S(t)/T(t)$.

The response to radiative forcing was estimated as a fourth-order polynomial fit to the precipitation change (%) of each model over the 1995–2100 period. The reference precipitation was determined using 1995–2014 means. Before regression fitting, the time series of precipitation change (1995–2100) was smoothed using the decadal running mean to reduce noise. Internal variability ($I$) was then calculated as the multi-model mean of the variance of the residuals from the fourth-order polynomial fits for a given model; internal variability does not change over time. Secondly, the model uncertainty ($M$) for each scenario was estimated from the variance of fitted values in different model simulations; model uncertainty components were considered to represent the multi-scenario mean. Lastly, scenario uncertainty ($S$) was calculated as the variance in the multi-model mean obtained for the three scenarios. For further details, refer to Hawkins and Sutton (2009, 2011).

**2.4 Moisture budget analysis**

To analyze the contribution of thermodynamic and dynamic mechanisms in future precipitation changes, we conducted a moisture budget analysis based on the linearized equation following previous studies (Seager and Naik 2012; Gao et al., 2012; Endo and Kitoh 2014; Li et al., 2015; Lee et al., 2017, 2018):

$$\delta P = \delta E + \delta TH + \delta DY + \delta NL + \text{res} \tag{2}$$

$$\delta TH = -\frac{1}{\rho_w g} \int_0^{p_s} \nabla \cdot (\bar{u}_{\text{CLIM}}[\delta \bar{q}]) \mathrm{d}p \tag{3}$$

$$\delta DY = -\frac{1}{\rho_w g} \int_0^{p_s} \nabla \cdot ([\delta \bar{u}]\bar{q}_{\text{CLIM}}) \mathrm{d}p \tag{4}$$

$$\delta NL = -\frac{1}{\rho_w g} \int_0^{p_s} \nabla \cdot (\delta \bar{u} \delta \bar{q}) \mathrm{d}p \tag{5}$$

$$\delta(\cdot) = (\cdot)_{\text{LT}} - (\cdot)_{\text{CLIM}} \tag{6}$$

where P is precipitation; E is surface evaporation; TH, DY, and NL are the thermodynamic, dynamic, and nonlinear terms, respectively; u is the horizontal wind vector; q is the specific humidity; p is the pressure; $\rho_w$ is the water density; g is the gravitational acceleration; and $p_s$ is the surface pressure. CLIM and LT indicate the climatology (1995–2014) and long-term periods (2081–2100), respectively. The unit of all terms is mm day-1. Overbars indicate climatological monthly means. To analyze future precipitation changes, we considered E, TH, DY, and NL, which were calculated using monthly mean values. Before the analysis, the evaporation, surface pressure, zonal wind, meridional wind, and specific humidity data were re-gridded

into a 2.5° × 2.5° resolution. Vertical integration of the TH, DY, and NL terms was conducted at six pressure levels: 1000, 850, 700, 500, 250, and 100 hPa.

## 3 Results

### 3.1 Evaluation and projection of regional inter-monthly precipitation patterns

Before analyzing future rainfall projections, we evaluated the climatology of inter-monthly precipitation patterns over East Asia. The CMIP6 and CMIP5 model ensembles generally reproduced the observed climatology patterns of the subseasonal rainband evolution in East Asia and its three subregions (Fig. 1). However, the two CMIPs tend to simulate weaker amplitudes than the two observations, particularly at the peak time. Nevertheless, the CMIP6 models simulated more intense precipitation than the CMIP5 models (Fig. 1 and Fig. S3).

In East Asian domain, the observed monsoon band moved northward from 20°N during June and reached 35°N in July. Both CMIP6 and CMIP5 faithfully reproduced this precipitation pattern during both the northward migration and the peak time of monsoon band. The CMIP6 ensemble captured the observed climatology more closely than the CMIP5 MME (Fig. S3). However, dry biases remained during the peak time. The location of the monsoon band was simulated well by both CMIPs.

In the China subregion, northward movement of the rain band occurred between 20°N and 32.5°N in June and approached 35°N in July (i.e., the "Mei-yu" frontal systems). The seasonal cycle of precipitation over China simulated by both the CMIP6 and CMIP5 MMEs was similar to that of the observed pattern, although the models underestimated the extent of northward migration of the monsoon band. The CMIP6 MME exhibited a better predictive performance of precipitation than CMIP5, but the dry biases remained in the monsoon band of the CMIP6 ensemble during peak time (Fig. S3). Additionally, 175    both CMIPs located the rain band further north than the observations.

    In Korea and Japan, the monsoon band moved northward from 25°N to 30°N in June and approached 40°N in July, as deduced from the GPCP and CMAP datasets (i.e., the frontal system termed "Changma" in Korea and "Baiu" in Japan). Both the CMIP5 and CMIP6 MMEs underestimated precipitation during the rainy season. Critically, neither model simulated a precipitation higher than that actually observed during the peak time in Korea, and only one model (MIROC6) simulated 180    more precipitation than what was observed during the northward migration of the monsoon band. Moreover, the CMIP6-modeled rain band reached a similar latitude to that of the observed band during the peak time in Korea. The dry bias of CMIP5 models was also reduced in CMIP6 models during the northward migration of the rain band over Japan (Fig. S3).

    The CMIP6 and CMIP5 MME future projections of zonal mean precipitation over the long term are displayed in Fig. 2. Precipitation is projected to increase along the monsoon band in all future scenarios. Stronger increases were recorded in 185    higher emission scenarios (SSP5-8.5 and RCP8.5). CMIP6 models generally simulated a stronger intensification in precipitation than CMIP5 models. Despite the increase in precipitation along the monsoon band, the location of the monsoon band remained unchanged over East Asia and the three subregions.

Overall, both CMIP5 and CMIP6 reproduced the observed seasonal cycle well, with dry biases in East Asia and the three subregions. Of the two, the CMIP6 model generally improved the precipitation simulation of the monsoon band, particularly in terms of its northward migration. The projected enhanced precipitation is consistent with the findings of previous studies (Chen et al., 2021; Jiang et al., 2020; Xin et al., 2020). Therein, the improved performance of the CMIP6 model was ascribed to its use of a higher spatial resolution (Chen et al., 2021; Jiang et al., 2020) and its expression of smaller SST biases over the Northwestern Pacific Ocean (Xin et al., 2020).

To illustrate the details of projected changes in precipitation over East Asia and its three subregions, Figure 3 presents the northward migration and peak time of the monsoon band for our selected three future periods. The CMIP6 models projected, using three SSP and RCP scenarios, that precipitation will increase during both the northward migration and peak time of the monsoon band over East Asia and its three subregions. Generally, the change in precipitation rate was dependent on different emission scenarios and the future period; precipitation was projected to increase appreciably under higher emission scenarios and over long-term periods. The uncertainty range was also the widest in the higher-emission scenario and over the long-term period.

Under SSP1-2.6, SSP2-4.5, and SSP5-8.5, precipitation was projected to increase over the long-term in our East Asian domain by 9.8% (4.6–15.0% for the ±1 standard deviation range), 10.4% (4.7–16.0%), and 16.4% (8.4–24.5%), respectively, during the northward migration of the monsoon band, and by 9.4% (0.9–18.0%), 9.7% (3.0–16.4%), and 12.9% (3.7–22.2%), respectively, during the peak time (Fig. 3, see Table 1 for detailed projection values). Compared to CMIP5, CMIP6 generally projected stronger precipitation increases, particularly under high-emission scenario. For both the northward migration and peak time of the monsoon, more than two-thirds of the models agreed on the positive sign of the precipitation change, indicating robust long-term projections. However, the inter-model variability differed significantly between CMIP5 and CMIP6. In CMIP6, peak time exhibited larger variability than that during the northward migration period. In contrast, CMIP5 showed higher variability for the northward migration than for the peak time. Additionally, uncertainty ranges were generally larger in CMIP5 models, particularly in lower-emission scenarios (RCP2.6 vs. SSP1-2.6), suggesting greater inter-model consistency in CMIP6 projections.

Over China, precipitation was enhanced from 11% to 20% during the northward migration of the monsoon band and from 9% to 13% during the peak time in the long-term period of three SSP scenarios (Fig. 3, Table 1). For Korea, CMIP6 models projected a robust precipitation increase of 8-16% for the northward migration and 12-16% for the peak time (Table 1). In Japan, precipitation was projected to increase from 12% to 18% during the northward migration of the rain band and from 9% to 13% for the peak time over the long term period (Table 1). Notably, in CMIP6, Korea exhibited consistent precipitation increases during both the northward migration and peak time, while China and Japan showed a stronger intensification during the northward migration. The uncertainty range for peak time was larger than that for the northward migration in all three subregions under CMIP6, whereas in CMIP5, uncertainty was larger for the northward migration across all scenarios. Overall, CMIP6 models simulated more intense precipitation changes than CMIP5 models under the equivalent radiative forcing scenario, except for Japan where RCP8.5 projected a more significant increase than SSP5-8.5. These

differences highlight the overall improved model consistency and stronger precipitation projection in CMIP6 models compared to CMIP5 models.

## 3.2 Uncertainty factors

To examine how the uncertainty inherent to precipitation projections differ between models, we investigated the changes in total uncertainty as portioned among its three components as previously identified: internal variability, model uncertainty, and scenario uncertainty. Figure 4 shows the total fraction of variance for the two CMIPs as obtained from the analysis of three different future time periods, indicating which components are the most important in future projection uncertainty. In the near-term period analysis, internal variability (CMIP6: 39.3% and 42.4%, CMIP5: 30.7% and 49.1% for

the northward migration and the peak time, respectively) and model uncertainty (CMIP6: 60% and 55.7%, CMIP5: 67.1% and 50.4%) were the dominant contributors to projection uncertainty over East Asia. The contribution of internal variability (CMIP6: 28.8% and 34.3%, CMIP5: 23.7% and 41.1%) decreased slightly, while that of model uncertainty (CMIP6: 70.9% and 64.9%, CMIP5: 72.8% and 56.6%) increased slightly for the mid-term period analysis. In the long-term period analysis, the contribution of model uncertainty (CMIP6: 63.8% and 74.7%, CMIP5: 76.7% and 63.8%) to total uncertainty was dominant,

with the internal variability contribution decreased (CMIP6: 17.4% and 20.7%, CMIP5: 12.5% and 22.8%) and the scenario uncertainty contribution increased slightly (CMIP6: 18.8% and  4.6%, CMIP5: 10.8% and 13.4%). The contributions of all three of these components to projection uncertainty were similar for projections encompassing either East Asia or the three subregions individually (Table S3).

        Model uncertainty was the dominant contributor to CMIP6 projections of precipitation change during both the

northward migration of the rain band and the peak time throughout the 21$^{st}$ century, which is consistent with analyses of global mean precipitation and monsoon land precipitation projections (Hawkins and Sutton 2011; Lehner et al., 2020; Zhou et al., 2020). However, in our study, the contribution of internal variability was crucial in the near-term period analysis, whereas that of scenario uncertainty remained small. The corresponding results when using CMIP5 showed a similar pattern. We also conducted an absolute uncertainty analysis to determine the magnitude of each component contributing to projection

uncertainty and the extent of their contribution (Fig. S4). CMIP5 shows a larger model uncertainty in terms of absolute and relative uncertainties in comparison to CMIP6.

## 3.3 Moisture budget analysis

        To explore the mechanisms of inter-monthly precipitation changes in future projections, we conducted a moisture

budget analysis. We investigated how thermodynamic and dynamical mechanisms influence future projections of the intra-seasonal evolution of monsoon rain band over East Asia and its subregions. Figure 5 illustrates the change in precipitation and moisture budget terms in a long-term period analysis using CMIP6 and CMIP5 for three climate change scenarios over East

Asia and its three subregions. Increase in precipitation during the northward migration of the rain band and the peak time were mainly associated with an increase in the thermodynamic term and a general decrease in the nonlinear term in both CMIP6

and CMIP5. Increased precipitation was associated with either a decrease or an increase in the dynamic term, depending on the region. The thermodynamic terms in both two periods were overall larger in CMIP6, except for the northward period over Korea and Japan, and the uncertainty ranges in each term were large in CMIP5 than in CMIP6. Therefore, we conclude that the thermodynamic term plays a more dominant role in the occurrence of intense precipitation (due to increased moisture availability) than the dynamic and nonlinear terms. These results are consistent with those of previous studies (Chen et al.,

2020; Endo and Kitoh 2014; Lee et al., 2017, 2018).

Figure 6 shows the relationship between the MME precipitation change and the thermodynamic term, which combines three climate change scenarios from the two CMIPs over a long-term analysis period. Considering the northward shift of the rain band, the relationship between the MME precipitation change and thermodynamic term was statistically significant at the 10% level (based on a *t*-test) over the MMEs, including all six scenarios over all regions, except for China and Korea during

the monsoon peak time. This again confirms that the thermodynamic term explains a large part of the scenario uncertainty in long-term period projections. To investigate the contribution of the dynamic term to inter-model uncertainty, we analysed the relationship between precipitation change and the dynamic term across models for the long-term period projections of precipitation change (Fig. 7). Precipitation change was significantly correlated with the dynamic term using three scenarios in the two CMIPs for all regions, except for the SSP2-4.5 and SSP5-8.5 scenarios over East Asia and the RCP8.5 scenario over

Korea during the peak time. This indicates that the dynamic term contributed to the model uncertainty in the long-term period projections, i.e. the spread of the CMIP6 and CMIP5 models in future precipitation projections was mainly caused by uncertainty in circulation changes.

To further examine regional differences among models, we analysed the inter-model regression patterns between future changes in the regional dynamic term and the 850hPa eddy geopotential height in peak time under the SSP2-4.5

scenarios for the period 2081–2100. The dynamic term over East Asia and China shows a strong correlation with the 850hPa eddy geopotential height (Fig. S5). While this relationship is statistically significant over East Asia (r=0.78) and China (r=0.46), no significant correlation is found over Korea and Japan (Fig. S5). This difference may be attributed to geographical contrasts between China and the Korea–Japan region, as well as differences in the timing of the northward progression of the monsoon rain band. When the WNPSH expands westward, enhanced moisture transport occurs over the South China Sea and southern

China, leading to increased rainfall over southern and central China (Huang et al., 2022; Zhou et al., 2020). In contrast, over Korea and Japan, the north-westward expansion of the WNPSH typically enhances moisture transport, thereby increasing precipitation. However, if the WNPSH extends excessively northward, the main rain band may shift into northern Japan, potentially reducing rainfall over Korea. This analysis provides insights into the regional contrast between China and the Korea–Japan region in terms of how the WNPSH influences precipitation patterns through dynamic processes. However, this

simply focused only on the WNPSH during the peak period under a single scenario (SSP2-4.5), and further investigation is needed to assess how other dynamic factors including SST patterns and upper-level circulations contribute to the inter-model

spread. In addition, future studies should consider multiple emission scenarios and intra-seasonal phases to better understand the robustness and variability of these regional differences.

**3.4 Thermodynamic and dynamic mechanisms**

To investigate the physical processes influencing the thermodynamic and dynamic terms, we conducted an inter-model correlation analysis between future changes in these terms and changes in global mean surface temperature (GMST), low-level moisture and circulation over a long-term period, following Zhou et al (2020) and Huang et al (2022).

Results indicate that GMST is found to have a strong relationship with the thermodynamic term over East Asia (Fig. 8c,h). The inter-model correlation coefficient between future changes in thermodynamic term and GMST are 0.89 during northward migration and 0.79 during peak time, respectively. These indicate that greater increase in GMST lead to stronger precipitation responses driven by the thermodynamic term. To identify the SST region associated with the thermodynamic term while excluding the effect of global warming, we calculated the inter-model correlation pattern between the thermodynamic residual term and the 850hPa specific humidity (q850) residual using all three SSP scenarios (Fig. 8a,f). The residuals were obtained from the regression analysis of each variable against GMST on grid base. Because future changes in thermodynamic term are calculated using future changes in q850 (Eq. 3), the q850 residual was used as an intermediary variable to indirectly examine the influence of regional SST patterns. We then identified a region of high correlation between q850 residual and the thermodynamic residual term to the south of the East Asian domain for thermodynamic term (black box), and selected this area (purple box). The location of q850 to the south of the precipitation region is consistent with the convergence term in the moisture budget equation (Eq. 3). The correlation coefficients between the area–averaged q850 residual and the thermodynamic residual term are 0.40 for northward migration and 0.36 for peak time, both of which are statistically significant (Fig. 8d,i). This confirms that the area–averaged q850 residual effectively explains the inter-model spread of the thermodynamic residual term.

Next, we computed the correlation pattern between the area–averaged q850 residual and the SST residual (Fig. 8b,g) and identified regions with high correlation coefficients. During the northward migration, area–averaged q850 residual is strongly correlated with the SST residual over the western North Pacific (WNP; 130–190°E, 10–25°N; purple box in Fig. 8b), with a correlation coefficient of 0.63, which is statistically significant at the 5% level (Fig. 8e). This indicates that the SST warming over the WNP enhances local evaporation, increasing moisture availability for transport into East Asia, and thereby contributing to precipitation increases driven by the thermodynamic residual term. During peak time, the area–averaged q850 residual is significantly correlated with the SST residual over the tropical central Pacific (TCP; 160°E–150°W, 10–25°N; purple box in Fig. 8g), with a correlation coefficient of 0.32 (Fig. 8j). This implies that SST changes in the TCP may modulate q850 residual variability, which could in turn influence thermodynamically driven precipitation changes over East Asia.

The results show that inter-model spread in the thermodynamic term is primarily driven by global warming, accounting for approximately 80% and 67% of the variance during the northward migration and peak time, respectively. In contrast, regional SST patterns contribute only marginally to the thermodynamic term, exerting their influence indirectly through q850 residuals and ultimately accounting for only small portions of thermodynamic term spread during both periods. Therefore, it is likely that global warming dominates the inter-model spread in the thermodynamic term over East Asia while contribution from regional SST patterns is minimal.

In contrast to the thermodynamic term, the inter-model spread of the dynamic term exhibits near zero correlation with GMST (Fig. 9c,h). To investigate the factors driving the inter–model spread of dynamic term, we examined 850hPa relative vorticity, which is a key factor controlling the dynamic term. Figure 9a and f show the inter-model correlation patterns between dynamic residual term and the 850hPa relative vorticity residual. Strong correlations with negative vorticity were identified in the region south of the East Asian precipitation domain (i.e. the area used to compute the regional mean dynamic term; black box in Fig. 9a,f). Based on this, we calculated the area–averaged relative vorticity over the regions showing the highest correlation: 100-150°E, 10-25°N for the northward migration and 100-150°E, 15-35°N for the peak time (purple box in Fig. 9a,f).

During the northward migration, relative vorticity exhibits a strong negative correlation with the dynamic residual term (r=-0.60, Fig. 9d), indicating that enhanced anti-cyclonic circulation over this region is associated with increased precipitation driven by dynamic term in East Asia. This anti-cyclonic circulation appears to be closely linked to the dipole SST pattern between the Bay of Bengal and South China Sea (BOB-SCS; 80-120°E, 0-20°N) and the tropical western Pacific (TWP; 140°E-160°W, 0-20°N; purple boxes in Fig. 9b). Figure 9e shows a scatter plot of relative vorticity against the residual SST difference between the BOB-SCS and TWP. The correlation is highly significant (r = -0.64), indicating that a large positive SST gradient between the two regions strengthens anti-cyclonic circulation, enhances moisture transport into East Asia, and induces increased precipitation. This result is consistent with the CMIP5-based analysis by He and Zhou (2015). During the peak time, the correlation between the dynamic term and the area-averaged relative vorticity remains significant (r=-0.56, Fig. 9i). The relative vorticity is strongly correlated with SST warming over mid-latitude North Pacific warming (130-180°E, 20-40°N), with a correlation coefficient of -0.56. Additionally, the SST difference between the BOB-SCS and TWP remains significantly correlated with relative vorticity at peak time (r = -0.45) as shown in the SST residual correlation pattern (Fig. 9g). These results suggest that, during peak time, the inter-model spread of the dynamic term is strongly associated with negative relative vorticity (i.e., anti-cyclonic circulation) over the Northwest Pacific. This negative vorticity is closely linked to both mid-latitude North Pacific SST warming and the tropical SST gradient.

These results indicate that the inter-model spread of the dynamic term is primarily linked to variation in low-level circulation rather than global warming. In particular, negative relative vorticity over the western North Pacific is closely associated with tropical SST gradient between the BOB-SCS and TWP (northward migration) and mid-latitude North Pacific warming. These factors contribute to the model uncertainty in East Asia precipitation projections driven by the dynamic term. It should be noted that our brief examination focused on identifying the SST regions that can explain the inter-model spread

of the thermodynamic and dynamic terms in separation from global warming influences. More comprehensive analysis of the underlying physical mechanisms is warranted to explore the SST-related processes in the identified regions.

**4 Summary and conclusion**

To quantify the inter-monthly evolution of monsoon rain band in East Asia and three subregions, we defined two indices based on (1) the northward migration of the monsoon band and (2) the peak time of the monsoon. Using this metric, we evaluated and compared projections in precipitation change between CMIP6 and CMIP5 models of various future emission scenarios. We also analysed the sources of projection uncertainty under three SSP scenarios for three future periods (near-term, mid-term, and long-term) and determined the relative contributions of  the thermodynamic  and dynamic terms to the
uncertainty based on a moisture budget analysis.

        The CMIP6 models reproduced the observed climatology patterns of regional inter-monthly evolution of monsoon rain band in precipitation well, and the overall performance of the CMIP6 MME was improved compared to that of the CMIP5 MME. However, dry bias remained in the CMIP6 MME. The models projected that precipitation would exhibit an overall increase during both the northward migration of the rain band and during peak time over East Asia and the three subregions.
This projected increase in precipitation was greater for the period of northward migration than for the peak time over all regions, except for Korea. However, inter-model variability was greater during the peak time than during the northward migration of the rain band. Intense precipitation was projected over the long term and was associated with thermodynamic responses due to increased moisture availability. Model uncertainty and internal variability were the main contributors to the total uncertainty of precipitation projections. For long-term projections, scenario uncertainty accounted for approximately 10% of the total
variance. The inter-model variability in precipitation change was mainly caused by differences in dynamic term resulting from circulation changes across the models while thermodynamic term was significantly associated with scenario uncertainty.

        Through inter-model correlation analysis, we have further shown that the scenario-dependent precipitation changes driven by thermodynamic terms are largely explained by global warming with a small contribution by regional SST. In contrast, the large diversity of dynamic terms is not affected by global warming but by regional SST warming patterns and corresponding
expansion and northward shift of the western North Pacific subtropical high, shaping moisture transport to the target subregion during different subseasons. Thus, further analysis is needed to identify detailed physical processes behind the diverse dynamic effects on regional inter-monthly precipitation patterns, as these factors will be critical in quantifying uncertainties in future precipitation projections.

        Our results for the CMIP6 model performance and future projections were consistent with those of previous global
and Asian monsoon precipitation studies based on different metrics (Chen et al., 2020, 2021; Moon and Ha 2020; Xin et al., 2020). In terms of regional and inter-monthly variations in precipitation, the simulation performance of CMIP6 models was superior to that of CMIP5 and CMIP3 (Kusunoki and Arakawa 2015). This may be attributable to the higher spatial resolution and improved model physics of CMIP6 (cf. Eyring et al., 2019; Paik et al., 2020). To confirm the spatial resolution effect,

future research can utilize the multi-tiered HighResMIP subproject of CMIP6, which enables the systematic investigation of

the resolution impact for the past and future climate (Haarsma et al., 2016). Some studies reported that increased spatial resolution did not change model skill metrics appreciably, depending on regions (e.g., Xin et al., 2021; Wehenr et al., 2021), which indicates the importance of assessing physical processes associated with the intra-seasonal evolution of monsoon rain band.

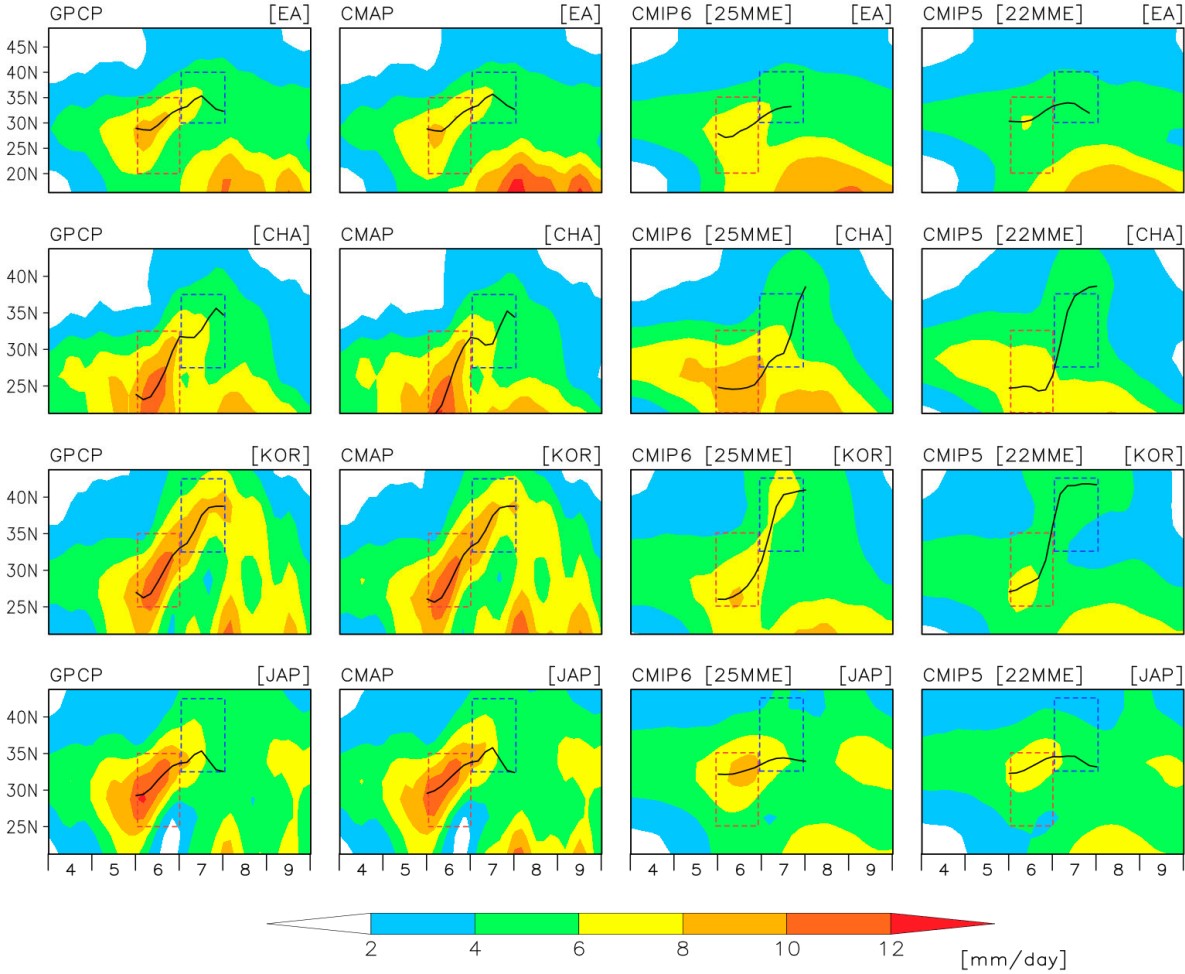


**Figure 1. Time–latitude cross sections of pentad precipitation rates (mm/day) from 1995 to 2014 over East Asia (EA; 100–150°E), China (CHA; 110–120°E), Korea (KOR; 120–130°E), and Japan (JAP; 130–142°E) from the Global Precipitation Climatology Project (GPCP), Climate Prediction Center Merged Analysis of Precipitation (CMAP), Coupled Model Intercomparison Project Phase 6 (CMIP6) Multi-Model Ensemble (MME), and CMIP5 MME. Red and blue boxes represent the northward migration and**
**peak time of monsoon rain bands, respectively. Black lines follow the latitude with the precipitation maximum (dPR/dlat = 0). The red and blue boxes indicate the timing and latitude of northward migration and peak time, respectively.**

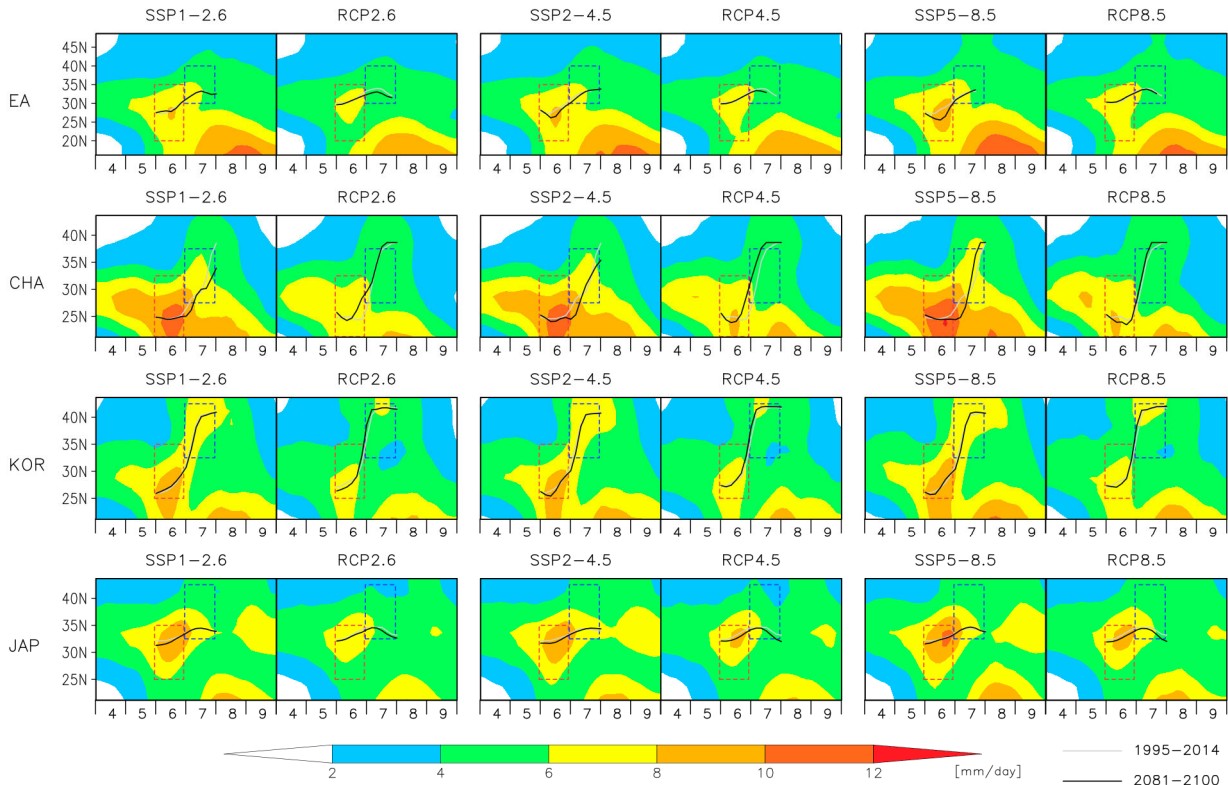

Figure 2. Time–latitude cross sections of pentad precipitation (mm/day) over East Asia (EA), China (CHA), Korea (KOR), and Japan (JAP) in a long-term period (2081–2100) from six future scenarios : Shared Socioeconomic Pathway (SSP)1-2.6, Representative Concentration Pathway (RCP)2.6, SSP2-4.5, RCP4.5, SSP5-8.5, and RCP8.5. Red and blue boxes represent the northward migration and peak time of monsoon rain bands, respectively. The gray and black lines follow latitudes with the precipitation maximum in 1995–2014 and in 2081–2100, respectively. The red and blue boxes indicate the timing and latitude of northward migration and peak time, respectively.

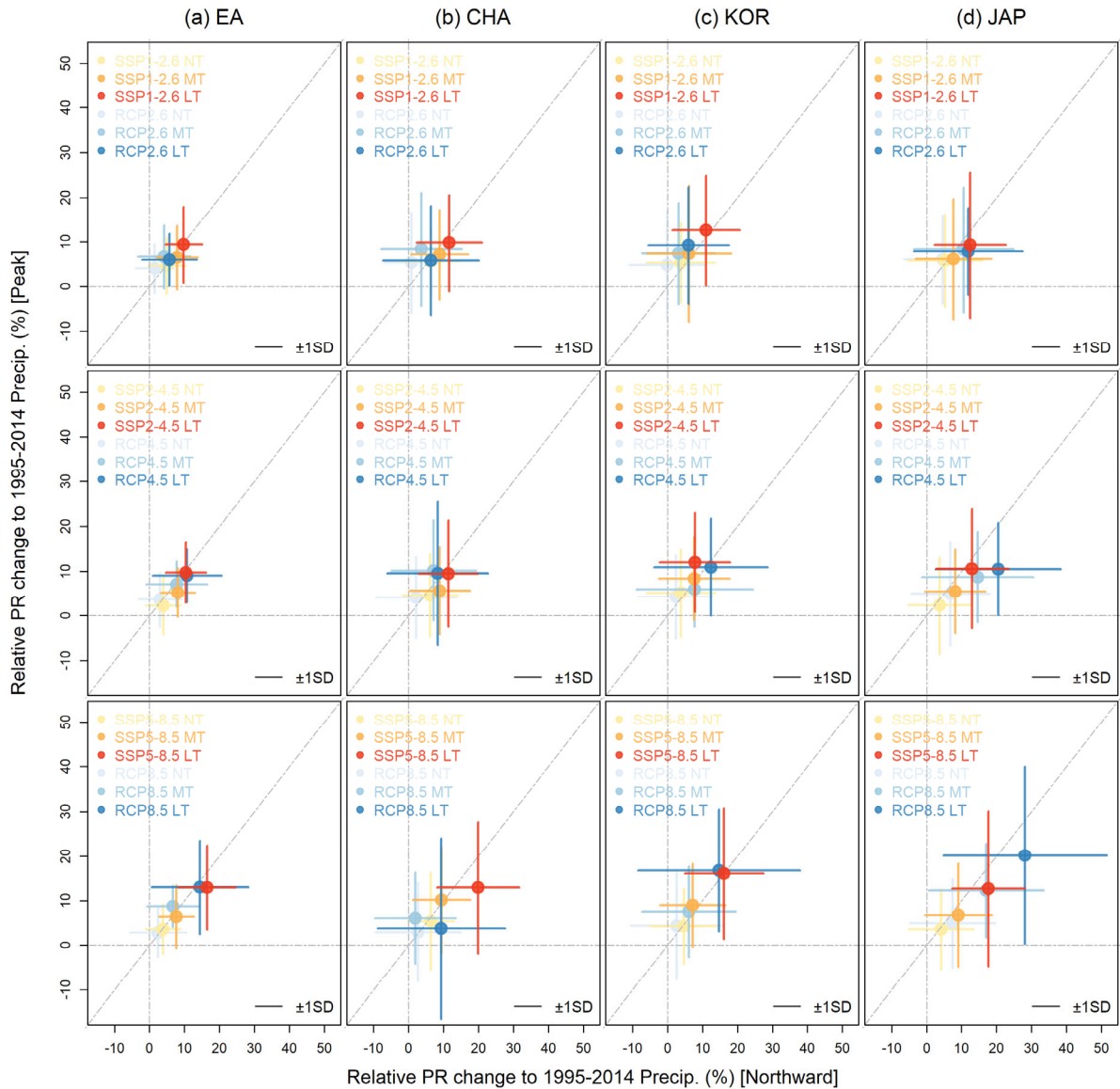

Figure 3. Scatter plot of changes in precipitation (PR) during the northward migration (x-axis) and peak time (y-axis) of the monsoon band over (a) East Asia (EA), (b) China (CHA), (c) Korea (KOR), and (d) Japan (JAP) for near-term (NT; 2021–2040), mid-term (MT; 2041–2060), and long-term (LT; 2081–2100)) from six future scenarios: Shared Socioeconomic Pathway (SSP)1-2.6 and Representative Concentration Pathway (RCP) 2.6 (1st row), SSP2-4.5 and RCP4.5 (2nd row), and SSP5-8.5 and RCP8.5 (3rd row). Projections are based on the climatological data during 1995–2014. Error bars indicate ±1 standard deviation (SD) for values calculated via the Coupled Model Intercomparison Project Phase 6 and Phase 5 models. The gray horizontal and vertical dashed lines indicate the zero percentage of relative precipitation change to climatology (1995-2014 mean).

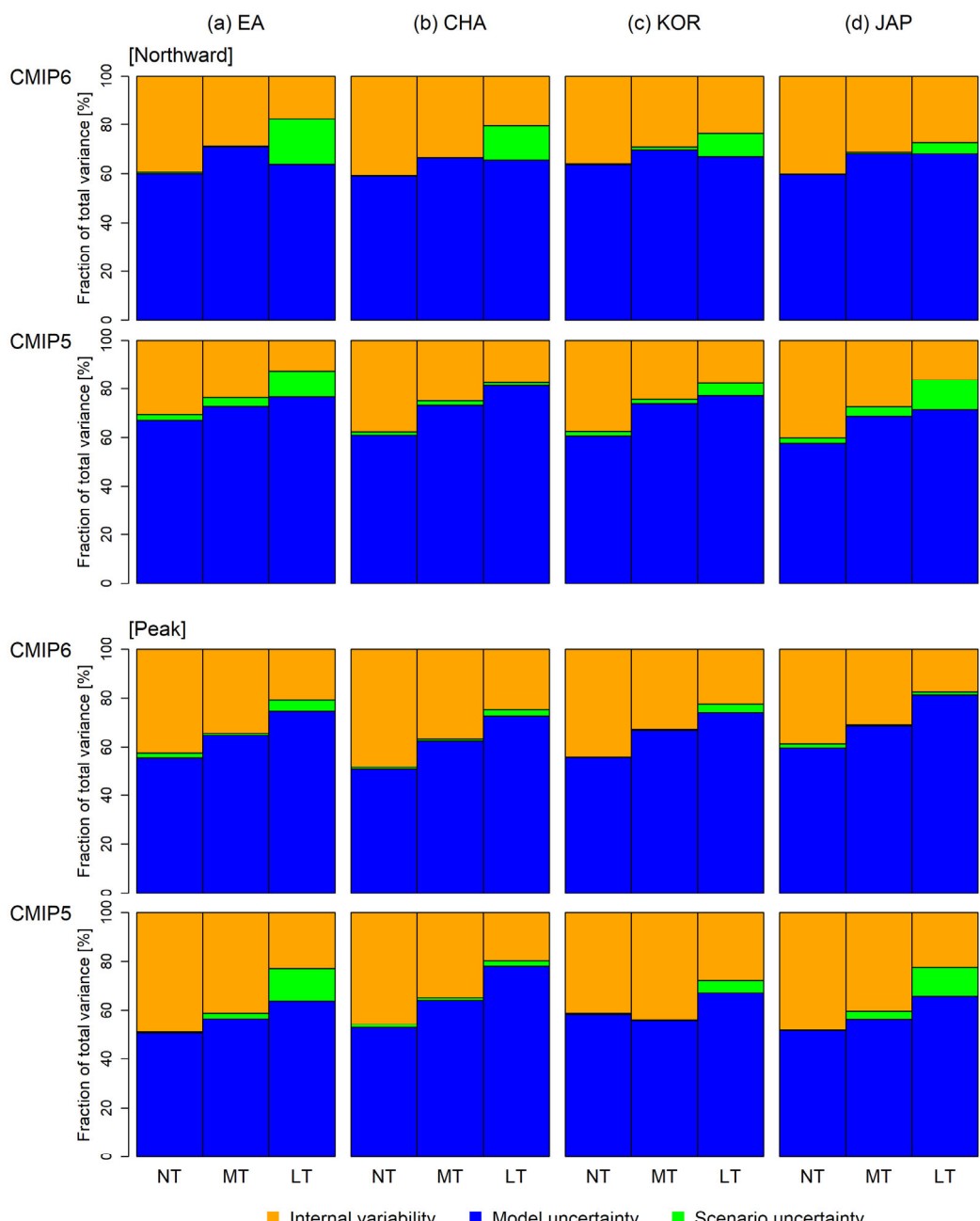

**Figure 4. Relative contributions (%) of internal variability (orange), model uncertainty (blue), and scenario uncertainty (green) to variance in total projection uncertainty for near-term (NT), mid-term (MT), and long-term (LT) of precipitation change over (a) East Asia (EA), (b) China (CHA), (c) Korea, and (d) Japan (JAP) for the northward migration (upper two panels) and the peak time (bottom two panels) of the monsoon band. The top panel in each of these two sets illustrates results obtained via CMIP6, while the bottom panel contains results obtained with CMIP5.**


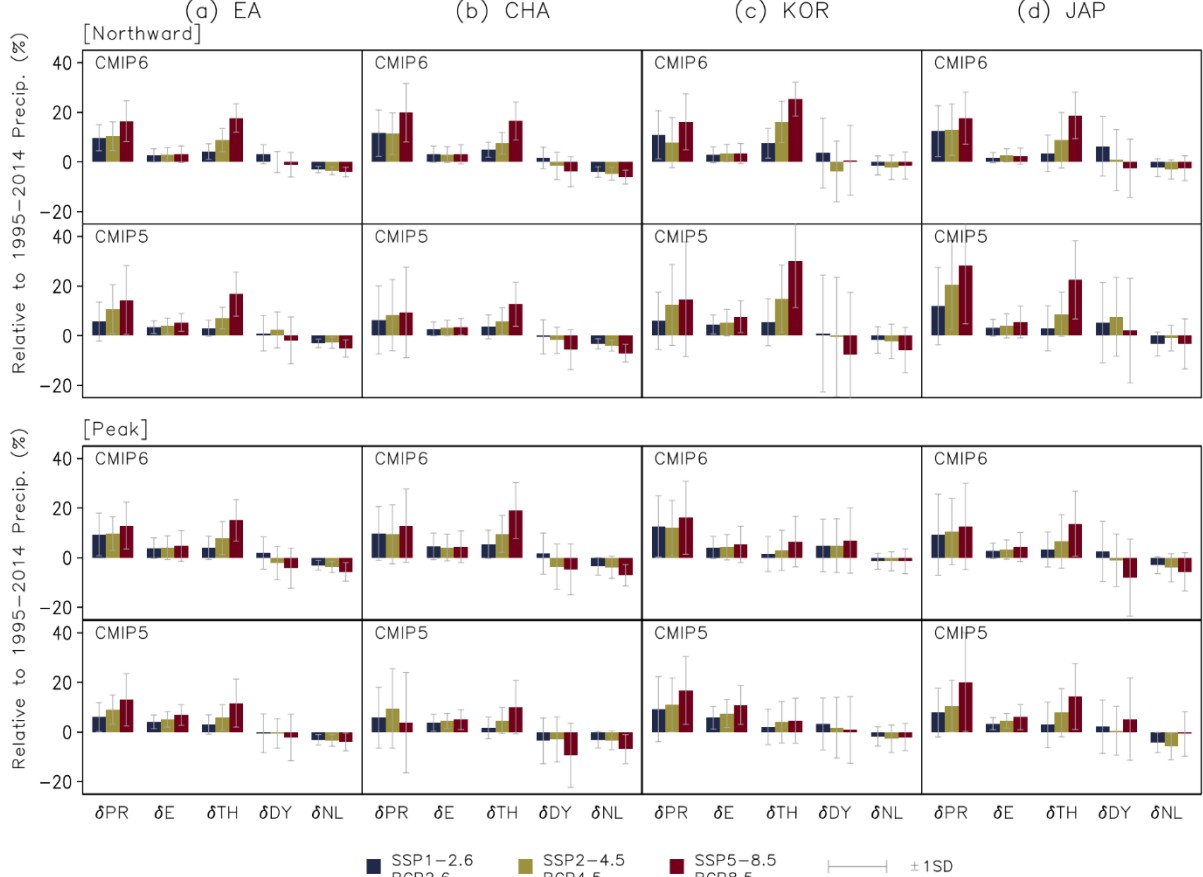

**Figure 5.** Change in moisture budget terms—precipitation (PR), evaporation (E), thermodynamic (TH), dynamic (DY), and nonlinear (NL)—for long-term (2081–2100) analyses over (a) East Asia (EA), (b) China (CHA), (c) Korea (KOR), and (d) Japan (JAP), relative to the climatology during 1995–2014 from six future scenarios: Shared Socioeconomic Pathway (SSP)1-2.6 and Representative Concentration Pathway (RCP)2.6 (blue bars), SSP2-4.5 and RCP4.5 (green bars), and SSP5-8.5 and RCP8.5 (dark red bars). Gray error bars indicate the ±1 standard deviation (SD) range of the models.


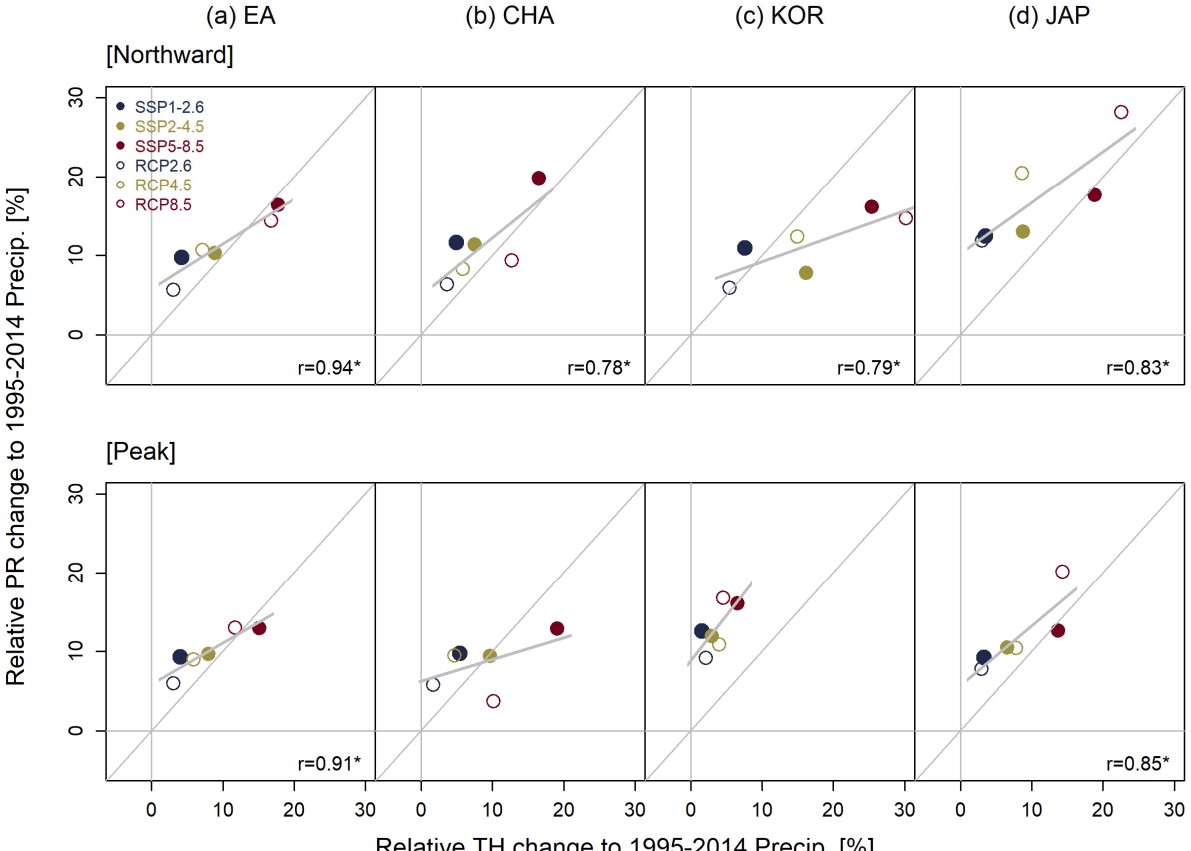

Figure 6. Scatter plot of the mean multi-model ensemble (MME) precipitation change (PR) and thermodynamic (TH) terms, with three climate change scenarios being used in each of the CMIP6 (closed circles) and CMIP5 (open circles) models for long-term (2081–2100) analyses covering (a) East Asia (EA), (b) China (CHA), (c) Korea (KOR), and (d) Japan (JAP). Projection were calculated relative to the climatology precipitation data from 1995–2014. Gray lines represent the linear regression slopes (with statistical significance at the 10% level). Different circle formats represent results obtained using the following climate change scenarios: Shared Socioeconomic Pathway (SSP)1-2.6 (closed blue circle), SSP2-4.5 (closed green circle), SSP5-8.5 (closed red circle), Representative Concentration Pathway (RCP)2.6 (open blue circle), RCP4.5 (open green circle), and RCP8.5 (open red circle).

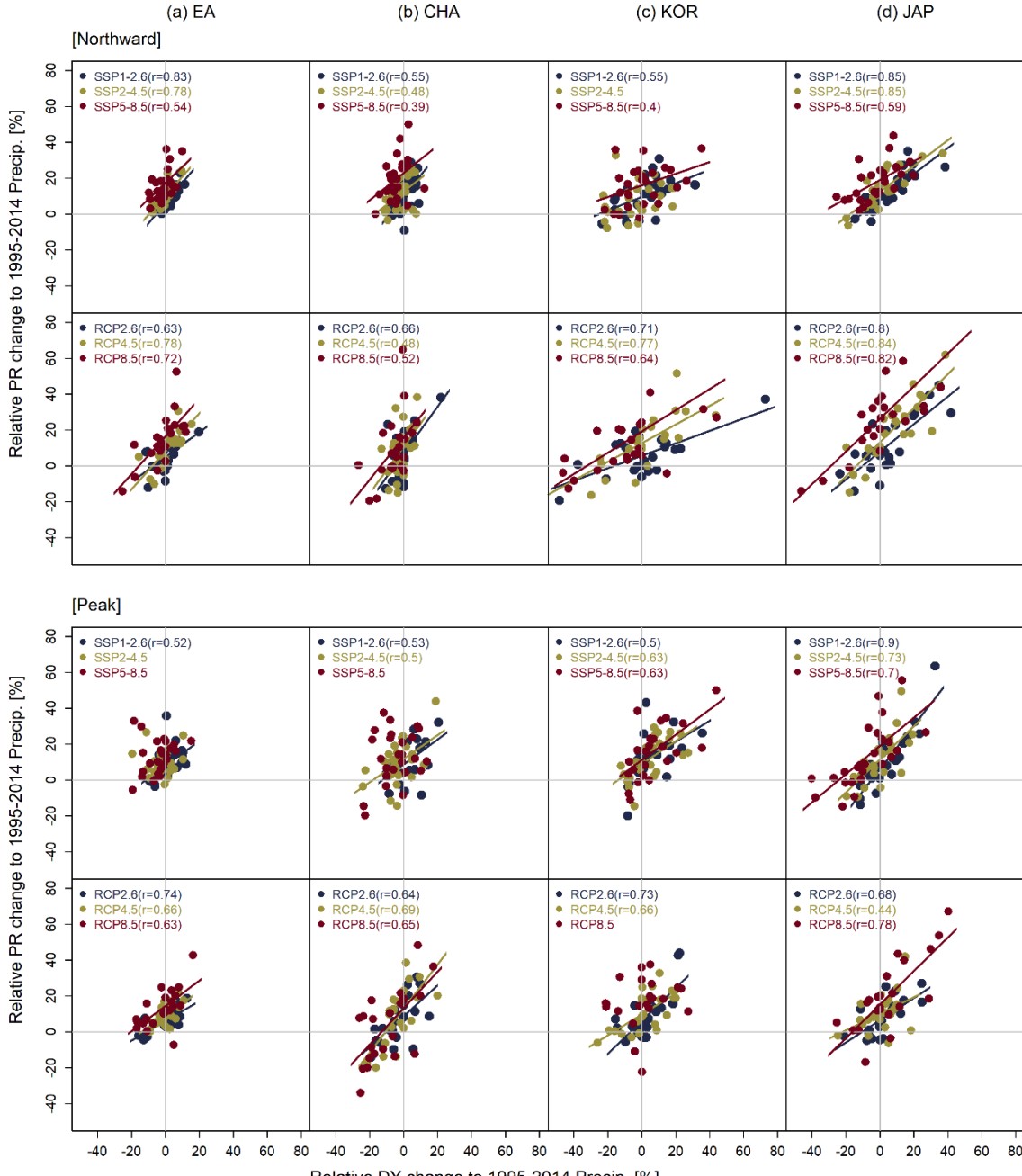

**Figure 7. Scatter plot of precipitation change (PR) and dynamic (DY) terms for long-term (2081-2100) analyses covering (a) East Asia (EA), (b) China (CHA), (c) Korea (KOR), and (d) Japan (JAP). Future changes were calculated relative to the 1995–2014. Colored lines represent the linear regression slopes (with statistical significance at the 10% level), along with their associated correlation coefficients (r).**

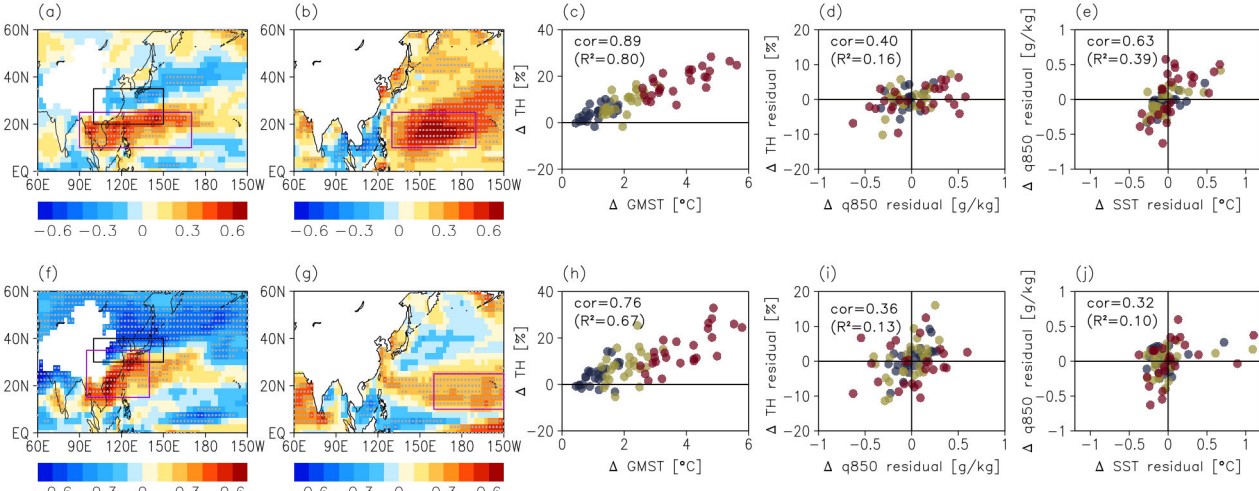


**Figure 8. Analysis of global warming and regional SST influence on thermodynamic changes (TH) for the (a-e) northward migration and peak time (f-j). (a, f) Inter-model correlation patterns between the TH residual term and the 850hPa specific humidity (q850) residual. The black boxes indicate the East Asia region used for calculating the TH term. The purple boxes represent regions with high correlation between the TH residual and q850 residual (90–170°E, 10–25°N in a and 90–140°E, 15–35°N in f). (b, g) Inter-model**

**correlation patterns between the area-averaged q850 residual (purple box from a and f) and the SST residual, with purple boxes highlighting SST regions that are strongly correlation with the area-averaged q850 residual. Scatter plots of (c, h) GMST vs. TH, (d, i) q850 residual vs. TH residual, and (e, j) SST residual vs. q850 residual. Navy, olive green, and dark red dots indicate SSP1-2.6, SSP2-4.5, and SSP5-8.5, respectively. Correlation coefficients are based on all models across scenarios. Refer to the main text for the detailed latitude-longitude range of the selected regions (black and purple boxes).**


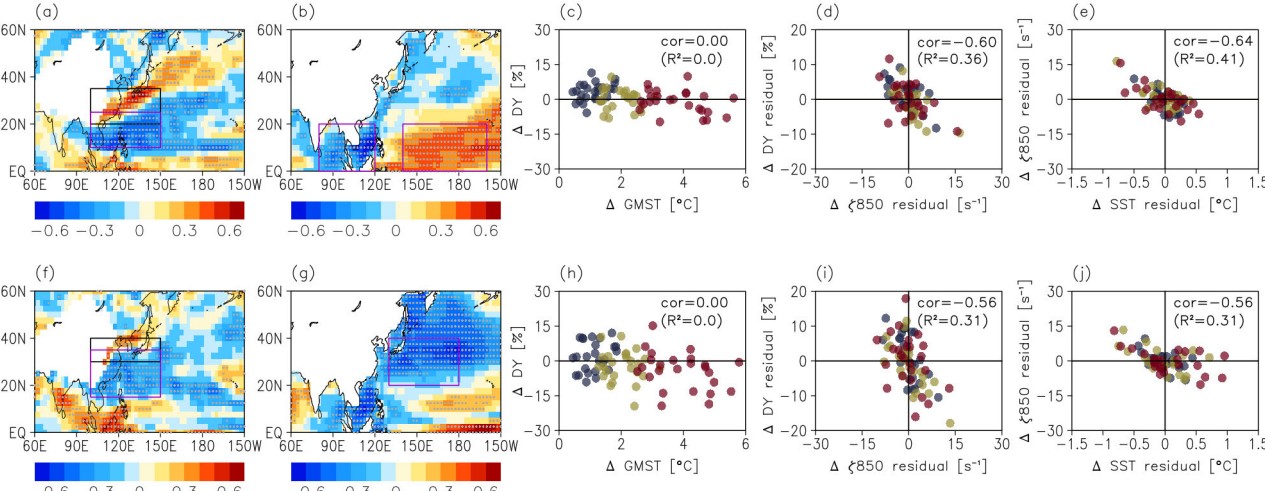

**Figure 9. Analysis of global warming and regional SST influence on dymanic changes (DY) for the (a-e) northward migration and peak time (f-j). (a, f) Inter-model correlation patterns between the DY residual term and the 850hPa relative vorticity (ζ850) residual. The black boxes indicate the East Asia region used for calculating the DY term. The purple boxes represent regions with high correlation between the DY residual and ζ850 residual. (b, g) Inter-model correlation patterns between the area-averaged ζ850 residual (purple box from a and f) and the SST residual, with purple boxes highlighting SST regions that are strongly correlation with the area-averaged ζ850 residual. Scatter plots of (c, h) GMST vs. DY, (d, i) ζ850 residual vs. DY residual, and (e, j) SST residual vs. ζ850 residual. In (e), the SST residual represents the SST difference between the two tropical ocean regions in b. Navy, olive green, and dark red dots indicate SSP1-2.6, SSP2-4.5, and SSP5-8.5, respectively. Correlation coefficients are based on all models across scenarios. Refer to the main text for the detailed latitude-longitude range of the selected regions (black and purple boxes).**



**Table 1. Future precipitation changes in the long-term period (2081-2100) compared to the period 1995-2014 in East Asia (EA), China(CHA), Korea(KOR), and Japan(JAP) during the northward migration and peak time under SSP and RCP scenarios.**

|  |  | Northward migration | Peak time |
|---|---|---|---|
| EA | SSP1-2.6 | 9.8%(4.6~15.0%) | 9.4%(0.9~18.0%) |
|  | SSP2-4.5 | 10.4%(4.7~16.0%) | 9.7%(3.0~16.4%) |
|  | SSP5-8.5 | 16.4%(8.4~24.5%) | 12.9%(3.7~22.2%) |
|  | RCP2.6 | 5.7%(-1.9~13.3%) | 6.0%( 0.3~11.7%) |
|  | RCP4.5 | 10.7%(1.1~ 20.3%) | 9.0%( 3.3~14.8%) |
|  | RCP8.5 | 14.4%(0.9~27.9%) | 13.0%( 2.7~23.3%) |
| CHA | SSP1-2.6 | 11.6%(2.5~20.8%) | 9.8%(-0.8~20.4%) |
|  | SSP2-4.5 | 11.4%(3.1~19.6%) | 9.4%(-2.2~21.1%) |
|  | SSP5-8.5 | 19.9%(8.3~31.4%) | 12.9%(-1.5~27.4%) |
|  | RCP2.6 | 6.4%(-7.0~19.8%) | 5.8%(-6.2~17.8%) |
|  | RCP4.5 | 8.3%(-5.7~22.4%) | 9.5%(-6.1~25.1%) |
|  | RCP8.5 | 9.4%(-8.4~27.2%) | 3.8%(-16.0~23.6%) |
| KOR | SSP1-2.6 | 11.0%(1.5~20.5%) | 12.6%(0.5~24.7%) |
|  | SSP2-4.5 | 7.8%(-2.0~17.7%) | 12.0%(1.1~22.8%) |
|  | SSP5-8.5 | 16.1%(5.1~27.2%) | 16.1%(1.7~30.5%) |
|  | RCP2.6 | 6.0%(-5.3~17.3%) | 9.2%(-3.6~22.0%) |
|  | RCP4.5 | 12.4%(-3.5~28.3%) | 10.9%(0.3~21.5%) |
|  | RCP8.5 | 14.7%(-7.9~37.3%) | 16.8%(3.4~30.2%) |
| JAP | SSP1-2.6 | 12.4%(2.4~22.5%) | 9.3%(-6.8~25.3%) |
|  | SSP2-4.5 | 13.0%(2.9~23.2%) | 10.6%(-2.5~23.6%) |
|  | SSP5-8.5 | 17.7%(7.4~27.9%) | 12.7%(-4.3~29.7%) |
|  | RCP2.6 | 11.9%(-3.3~27.2%) | 7.9%(-1.7~17.4%) |
|  | RCP4.5 | 20.5%(3.0~38.0%) | 10.5%(0.3~20.6%) |
|  | RCP8.5 | 28.2%(5.3~51.0%) | 20.1%(0.7~39.6%) |


**Data availability**

The utilized CMIP5 and CMIP6 model simulations are listed in table s1 and table s2. All of the datasets can be accessed at https://esgf-node.llnl.gov/search/cmip5/ and https://esgf-node.llnl.gov/search/cmip6/, respectively.


**Author contributions**

SKM conceived the study; YHK conducted the analysis and wrote the manuscript draft; SKM reviewed and edited the manuscript.

**Competing interests**

The authors declare that they have no conflict of interest.

**Acknowledgements**

This study was supported by the Korea Meteorological Administration Research and Development Program under Grant RS-2024-00403386 and the Human Resource Program for Sustainable Environment in the 4th Industrial Revolution Society.

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
