# Peer review of "Future changes in regional inter-monthly precipitation patterns of the East Asian summer monsoon and associated uncertainty factors"

_Earth System Dynamics, 2024_

## Author Comment (AC1)

RC1

This study investigates the future changes in spatiotemporal precipitation patterns of the East Asian Summer Monsoon (EASM) using CMIP6 models. It evaluates model performance, assesses future projections, and analyzes uncertainty factors. This study is meaningful but lacks more in-depth discussions on certain aspects. It is recommended that these areas be revised and clarified prior to publication.

Major comments:

1, The object of this study is the spatiotemporal precipitation patterns in East Asian. The two metrics (the time of northward movement of the monsoon band and peak of the monsoon band) might partly describe the spatiotemporal precipitation patterns. Did authors compare to other metrics or choose other methods?

Thank you for the insightful point. We will provide more detailed explanations on the precipitation indices in comparison with other East Asian monsoon circulation and timing indices as appropriate (e.g., Wang et al. 2008; Ha et al. 2020). We will also highlight the advantage of our indices in terms of representing the intraseasonal evolution of monsoon rain bands for East Asia and its three subregions by conducting further observational analyses using pentad precipitation data.

Wang, B., Wu, Z, Li, J., Liu, J., Chang, C.-P., Ding, Y., & Wu, G. (2008) How to measure the strength of the East Asia summer monsoon. Journal of Climate, 21, 4449- 4463.

Ha, K.-J., Moon, S., Timmermann, A. & Kim, D. (2020) Future changes of summer monsoon characteristics and evaporative demand over Asia in CMIP6 simulations. Geophys. Res. Lett. 47, e2020GL087492.

2, I would like to know, in the context of Moisture Budget Analysis, which processes influence the thermodynamic and dynamic terms—are they driven by global warming or regional sea surface temperature changes?

Thank you for the useful comment. Agreeing that identifying processes associated with the thermodynamic (TH) and dynamic (DY) terms is important, we will improve our interpretations of the physical mechanisms associated with the future precipitation change patterns. We will add more discussions on TH and DY effects by analyzing their relation with global warming and regional SST changes. Specifically, we will perform an inter-model regression analysis, following Zhou et al. (2020) and Huang et al. (2022). By regressing the model TH and DY terms against the low-level circulation and SST changes in the future, we will try to identify modeling factors associated with the inter-model spread in TH and DY in East Asia and provide associated discussions in the manuscript.

Zhou, S., Huang, G., & Huang, P. (2020). Inter-model spread of the changes in the East Asian summer monsoon system in CMIP5/6 models. Journal of Geophysical Research: Atmospheres, 125, 2020JD033016.

Huang, D., Liu, A., Zheng, Y., & Zhu, J. (2022). Inter-Model Spread of the Simulated East Asian Summer Monsoon Rainfall and the Associated Atmospheric Circulations From the CMIP6 Models. Journal of Geophysical Research: Atmospheres, 127, e2022JD037371.

3, Additionally, I suggest the authors provide a more detailed analysis of regional differences and their causes. For instance, it seems to me that compared to the other two regions, the models perform less effectively in simulating precipitation over China (Huang et al., 2022; Wang et al., 2020).

Reference:

Huang, D., Liu, A., Zheng, Y., & Zhu, J. (2022). Inter-Model Spread of the Simulated East Asian Summer Monsoon Rainfall and the Associated Atmospheric Circulations From the CMIP6 Models. *Journal of Geophysical Research: Atmospheres*, *127*, e2022JD037371. https://doi.org/10.1029/2022JD037371

Wang, B., Jin, C., & Liu, J. (2020). Understanding Future Change of Global Monsoons Projected by CMIP6 Models. *Journal of Climate*, *33*(15), 6471–6489. https://doi.org/10.1175/JCLI-D-19-0993.1

Thank you for the valuable suggestion. Agreeing that regional differences require more detailed analysis, we will compare inter-model spread across the three subregions and examine the potential causes of discrepancies across regions focusing on moisture budget analysis. For example, during the northward migration period, the inter-model differences in future precipitation changes are similar across all three subregions, but dynamic (DY) term exhibits a regional difference; i.e. smaller inter-model differences in China than the other two regions. We will try to identify factors (like atmospheric circulations and SST patterns) associated with the regional difference using composite and/or inter-model regression analyses.

4, Based on Fig.4, it indicates that the contribution of M to total uncertainty is dominant. What are the physical mechanisms for the model uncertainty is an important question. That means quantifying the model uncertainties and trace the sources are also important to understand the precipitation regime changes in the warming future. As I understand this is not the key focus of the paper, I think you could provide some hypotheses in the discussion section.

Thank you for the good point. We agree that providing discussions regarding the sources and mechanisms of model uncertainty would enhance the discussion and offer valuable context for future research. Building on our further analysis described above, we will outline key factors for model uncertainty and discuss their roles in shaping future changes in East Asia monsoon precipitation.

Minor comments

1, Line 61: predict→projection?

We will correct it as 'projection'.

2, Line 120, it is suggested to clarify the meaning of the subscript "REF" for better understanding.

"REF" refers to the climatology period, so we will change it to CLIM.

3, Authors have compared the results based on CMIP5 and CMIP6. However, it seems less differences between them.

They look similar in terms of future projections but we will further check detailed features for which CMIP6 models exhibit differences from CMIP5 models and discuss them in the revised manuscript.

4, Fig.1 ,Fig.2 , It should clarify how does red and blue boxes define.

We will revise the figure captions to better define those boxes.

5, Fig.3, What does the dashed lines mean?

We will explain the meaning of dashed lines (zero lines) in the figure caption.

---

## Author Comment (AC2)

RC2

The manuscript shows precipitation patterns in East Asia and three subregions using matrices. It compared projections in precipitation changes between CMIP5 and CMIP6 models of different SSP scenarios. The results are important, but the manuscript does not provide a physical basis for the conclusions. I suggest including the points below before publication.

The basis for choosing precipitation indices (1) the time of Northward migration of the rainband and (2) the peak of the monsoon band should be explained in detail although adopted from previous literature. What other monsoon indices were reported in previous studies? What is the advantage of choosing the above two monsoon indices should be given.

Thank you for the insightful point. We will provide more detailed explanations on the precipitation indices in comparison with other East Asian monsoon circulation and timing indices as appropriate (e.g., Wang et al. 2008; Ha et al. 2020). We will also highlight the advantage of our indices in terms of representing the intraseasonal evolution of monsoon rain bands for East Asia and its three subregions by conducting further observational analyses using pentad precipitation data.

Wang, B., Wu, Z, Li, J., Liu, J., Chang, C.-P., Ding, Y., & Wu, G. (2008) How to measure the strength of the East Asia summer monsoon. Journal of Climate, 21, 4449- 4463.

Ha, K.-J., Moon, S., Timmermann, A. & Kim, D. (2020) Future changes of summer monsoon characteristics and evaporative demand over Asia in CMIP6 simulations. Geophys. Res. Lett. 47, e2020GL087492.

The scientific basis for increased precipitation in different SSP scenarios and discussions on the associated physical mechanism is shallow.

Thank you for the useful comment. Based on your suggestions and questions provided below, we will improve our interpretations of the physical mechanisms associated with the future precipitation change patterns.

Discussions on thermodynamic effects on moisture thereby on precipitation enhancement should be elaborated.

We will add more discussions on thermodynamic effects by analyzing what thermodynamic factors in models. See below for details.

Influence of Wind and SST changes analysis should be provided to support the results.

We will perform an inter-model regression analysis, following Zhou et al. (2020) and Huang et al. (2022). By regressing the model TH (thermodynamic) and DY (dynamic) terms against the low-level wind and SST changes in the future, we will try to identify

modeling factors associated with the inter-model spread in TH and DY in East Asia and provide associated discussions in the manuscript.

Zhou, S., Huang, G., & Huang, P. (2020). Inter-model spread of the changes in the East Asian summer monsoon system in CMIP5/6 models. Journal of Geophysical Research: Atmospheres, 125, 2020JD033016.

Huang, D., Liu, A., Zheng, Y., & Zhu, J. (2022). Inter-Model Spread of the Simulated East Asian Summer Monsoon Rainfall and the Associated Atmospheric Circulations From the CMIP6 Models. Journal of Geophysical Research: Atmospheres, 127, e2022JD037371.

How does Hadley circulation change in different SSP scenarios?

Assuming that here you indicate the local Hadley cell in western North Pacific, we will check the possible role of local Hadley cell changes in shaping the East Asian monsoon precipitation patterns in different SSP scenarios.

Results on uncertainty factors do not quantify the uncertainty.

Thank you. We will provide a figure, so-called "fractional uncertainty", which shows actual values of the total uncertainty in precipitation changes and the relative contributions of internal variability, model uncertainty, and scenario uncertainty.

Precipitation values documents in section 3.1 should be tabulated

Thank you for the good suggestion. We will add a table showing the future precipitation changes in East Asia and its subregions during the northward migration and peak time.

Use of short forms M, I, TH …etc should be avoided although they are defined.

We will avoid some abbreviations for better reading throughout the manuscript.

---

## Author Comment (AC3)

RC3

I recommend that the manuscript undergo major revision before publication. Below are my comments:

Based on the title and the introduction, the paper aims to provide readers with a more comprehensive and detailed understanding of the changes in the spatiotemporal distribution of East Asian summer precipitation under warming scenarios. However, the results do not achieve this goal.

The spatiotemporal precipitation pattern is a key focus of the paper, but in this study, its primary role appears to be limited to the division into two time periods. This approach leads to the following issues:

What kind of change in the spatiotemporal precipitation pattern corresponds to the different changes in these two time periods? While indices can simplify the problem, the author's conclusions should focus on the precipitation pattern. However, in the abstract, the author merely states that precipitation increases across two periods and three regions. This undermines the significance of the indices painstakingly used throughout the paper.

Thank you for the clarifying comment. We agree that the chosen metrics provide only a partial perspective on the spatiotemporal precipitation patterns. These metrics were selected due to their ability to capture the space-time evolution of monsoon rain band, following previous studies (Kusunoki and Arakawa, 2015). However, it seems that the term "spatiotemporal precipitation pattern" has caused some confusion in understanding the metrics we used. This study focuses on the intraseasonal variations of precipitation along the movement of the monsoon band over the East Asia and its three subregions. Therefore, we will consider revising the term "spatiotemporal precipitation pattern" to "intra-seasonal evolution of monsoon rain band" or similar terms throughout the manuscript to better reflect our main results.

Does the authors suggest that the temporal evolution of East Asian summer monsoon precipitation can be sufficiently represented by averaging over just two time periods? If so, I think this point needs further evidence to support it.

Thank you for the good point. We agree that further evidence is needed for the representativeness of our precipitation indices for the temporal evolution of East Asian summer monsoon precipitation. First, we will provide more detailed explanations of our precipitation indices in comparison with other East Asian monsoon circulation and timing indices as appropriate (e.g., Wang et al. 2008; Ha et al. 2020). Secondly, we will highlight the advantage of our indices in representing the intraseasonal evolution of monsoon rain bands for East Asia and its three subregions by conducting further observational analyses using pentad precipitation data.

Wang, B., Wu, Z, Li, J., Liu, J., Chang, C.-P., Ding, Y., & Wu, G. (2008) How to measure the strength of the East Asia summer monsoon. Journal of Climate, 21, 4449- 4463.

Ha, K.-J., Moon, S., Timmermann, A. & Kim, D. (2020) Future changes of summer monsoon characteristics and evaporative demand over Asia in CMIP6 simulations. Geophys. Res. Lett. 47, e2020GL087492.

L45-55: Regarding the uncertainties in the East Asian monsoon projections, I suggest adding the following references:

Zhou S, Huang G, Huang P. A bias-corrected projection for the changes in East Asian summer monsoon rainfall under global warming. Climate Dynamics, 2019,54(1-2): 1-16.

Zhou S, Huang G, Huang P. Changes in the East Asian summer monsoon rainfall under global warming: Moisture budget decompositions and the sources of uncertainty. Climate Dynamics, 2018, 51(4): 1363–1373.

Zhou S, Huang P, Huang G, et al. Leading source and constraint on the systematic spread of the changes in East Asian and western North Pacific summer monsoon. Environmental Research Letters, 2019, 14(12): 124059.

Thank you for informing relevant references. We will include them in our revised manuscript to provide a more comprehensive discussion of the uncertainties in East Asian precipitation projections.

---

## Author Response (AR1)

**Reviewer 1:**

This study investigates the future changes in spatiotemporal precipitation patterns of the East Asian Summer Monsoon (EASM) using CMIP6 models. It evaluates model performance, assesses future projections, and analyzes uncertainty factors. This study is meaningful but lacks more in-depth discussions on certain aspects. It is recommended that these areas be revised and clarified prior to publication.

Thank you for your thoughtful comments. We have added more explanations and discussions following your suggestions in the revised manuscript. Please find our point-by-point responses below. Line numbers are based on the cleaned version of revised manuscript.

Major comments:

1, The object of this study is the spatiotemporal precipitation patterns in East Asian. The two metrics (the time of northward movement of the monsoon band and peak of the monsoon band) might partly describe the spatiotemporal precipitation patterns. Did authors compare to other metrics or choose other methods?

Thank you for the insightful point. We have provided more detailed explanations on the precipitation indices in comparison with other East Asian monsoon circulation indices. We have also highlighted the advantage of our indices in terms of representing the inter-monthly evolution of monsoon rain bands for East Asia and its three subregions by conducting further observational analyses as follows (**lines 100-125**).

"The developed indices were first evaluated in relation to East Asia precipitation patterns (Fig. S2a,b). Figure S1 shows the regression pattern of the northward migration and peak time index over East Asia and precipitation using GPCP data from 1995 to 2014. The regression patterns reveal the movement of the monsoon precipitation band during the northward migration and peak time, indicating that these indices are suitable for representing the inter-monthly evolution of monsoon rain band in East Asia.

For further evaluations of the indices, we examined the relationships between two precipitation-based indices and two East Asia summer monsoon indices: East Asia summer monsoon index [EASMI; defined as the difference between the 850hPa zonal wind anomalies averaged over the southern (100-150°E, 10-20°N) and northern (100-150°E, 25-35°N) regions; Zhang et al., 2003] and western North Pacific subtropical high [WNPSH; defined as the 850hPa eddy geopotential height averaged over 120°-150°E, 15-30°N; Zhou et al., 2020]. The EASMI shows a statistically significant negative correlation with the peak time index over East Asia (r=-0.49 for GPCP, r=-0.46 for GMAP) and with the northward migration index over China (r=-0.45 for GPCP). In contrast, the WNPSH exhibits a strong positive correlation with the northward migration index over China (r=0.63 for GPCP, r=0.50 for CMAP). However, their correlations with indices for Korea and Japan are generally weak and not statistically significant, suggesting that these circulation-based indices have limited ability to capture regional monsoon characteristics.

This is because these summer monsoon indices are based on the large-scale atmospheric circulation during the East Asia summer, and therefore have limitations in explaining regional rainfall mechanisms and intra-seasonal variability. Figure S2c and d show the regression patterns of the northward migration and peak time indices over East Asia with the 850hPa zonal wind, which is used to calculate the EASMI. During the peak time over East Asia, zonal wind anomalies in the two regions exhibit a strong correlation with the index, whereas no significant correlation is observed during the northward migration. Figure S2e and f show

the regression patterns of the northward migration and peak time indices over China with the 850hPa eddy geopotential height, which is used to calculate the WNPSH. The 850hPa eddy geopotential height shows a strong correlation during the northward migration over China but a weaker correlation during the peak time. Overall, our proposed indices for inter-monthly precipitation evolutions have the advantage of directly reflecting precipitation changes, better representing regional features, and allowing for quantitative analysis of the intra-seasonal evolution of monsoon rain band over East Asia."

In addition, considering your point and other reviewers' concerns, to better reflect our main findings, we have revised both the title and the terminology throughout the manuscript from "spatiotemporal precipitation pattern" to "regional inter-monthly precipitation pattern". Although the two metrics were selected based on previous studies (Kusunoki and Arakawa, 2015), we agree that the term "spatiotemporal precipitation pattern" can mislead the target of this study due to its broad meaning. We believe that "regional inter-monthly precipitation patterns" is more relevant and specific to represent what we analyze.

2, I would like to know, in the context of Moisture Budget Analysis, which processes influence the thermodynamic and dynamic terms—are they driven by global warming or regional sea surface temperature changes?

Thank you for the useful comment. To investigate the physical processes influencing the thermodynamic and dynamic terms, we conducted an inter-model correlation analysis between future changes in these terms and changes in global mean surface temperature (GMST), low-level moisture and circulation over a long-term period, following Zhou et al (2020) and Huang et al (2022). Results indicate that GMST is found to have a strong relationship with the thermodynamic term over East Asia while the inter-model spread of the dynamic term is primarily linked to variation in low-level circulation rather than global warming. We provided detailed results and associated discussions in the new subsection 3.4 Thermodynamic and Dynamic mechanism (**lines 290-353**) .

Zhou, S., Huang, G., & Huang, P. (2020). Inter-model spread of the changes in the East Asian summer monsoon system in CMIP5/6 models. Journal of Geophysical Research: Atmospheres, 125, 2020JD033016.

Huang, D., Liu, A., Zheng, Y., & Zhu, J. (2022). Inter-Model Spread of the Simulated East Asian Summer Monsoon Rainfall and the Associated Atmospheric Circulations From the CMIP6 Models. Journal of Geophysical Research: Atmospheres, 127, e2022JD037371.

3, Additionally, I suggest the authors provide a more detailed analysis of regional differences and their causes. For instance, it seems to me that compared to the other two regions, the models perform less effectively in simulating precipitation over China (Huang et al., 2022; Wang et al., 2020).

Reference:

Huang, D., Liu, A., Zheng, Y., & Zhu, J. (2022). Inter-Model Spread of the Simulated East Asian Summer Monsoon Rainfall and the Associated Atmospheric Circulations From the CMIP6 Models. *Journal of Geophysical Research: Atmospheres*, *127*, e2022JD037371. https://doi.org/10.1029/2022JD037371

Wang, B., Jin, C., & Liu, J. (2020). Understanding Future Change of Global Monsoons Projected by CMIP6 Models. *Journal of Climate*, *33*(15), 6471–6489. https://doi.org/10.1175/JCLI-D-19-0993.1

Thank you for the valuable suggestion. Agreeing that regional differences require more detailed analysis, we compared inter-model spread across the three subregions and examine the potential causes of discrepancies across regions focusing on moisture budget analysis. We have included the following discussion in the revised manuscript to clarify the regional difference under 3.3 Moisture budget analysis (**lines 273-288**).

"To further examine regional differences among models, we analysed the inter-model regression patterns between future changes in the regional dynamic term and the 850hPa eddy geopotential height in peak time under the SSP2-4.5 scenarios for the period 2081–2100. The dynamic term over East Asia and China shows a strong correlation with the 850hPa eddy geopotential height (Fig. S5). While this relationship is statistically significant over East Asia (r=0.78) and China (r=0.46), no significant correlation is found over Korea and Japan (Fig. S5). This difference may be attributed to geographical contrasts between China and the Korea–Japan region, as well as differences in the timing of the northward progression of the monsoon rain band. When the WNPSH expands westward, enhanced moisture transport occurs over the South China Sea and southern China, leading to increased rainfall over southern and central China (Huang et al., 2022; Zhou et al., 2020). In contrast, over Korea and Japan, the north-westward expansion of the WNPSH typically enhances moisture transport, thereby increasing precipitation. However, if the WNPSH extends excessively northward, the main rain band may shift into northern Japan, potentially reducing rainfall over Korea. This analysis provides insights into the regional contrast between China and the Korea–Japan region in terms of how the WNPSH influences precipitation patterns through dynamic processes. However, this simply focused only on the WNPSH during the peak period under a single scenario (SSP2-4.5), and further investigation is needed to assess how other dynamic factors including SST patterns and upper-level circulations contribute to the inter-model spread. In addition, future studies should consider multiple emission scenarios and intra-seasonal phases to better understand the robustness and variability of these regional differences."

4, Based on Fig.4, it indicates that the contribution of M to total uncertainty is dominant. What are the physical mechanisms for the model uncertainty is an important question. That means quantifying the model uncertainties and trace the sources are also important to understand the precipitation regime changes in the warming future. As I understand this is not the key focus of the paper, I think you could provide some hypotheses in the discussion section.

Thank you for the good point. We agree that providing discussions regarding the sources and mechanisms of model uncertainty would enhance the discussion and offer valuable context for future research. We have added a discussion in the summary and conclusion section as follow (**lines 372-378**):

"Through inter-model correlation analysis, we have further shown that the scenario-dependent precipitation changes driven by thermodynamic terms are largely explained by global warming with a small contribution by regional SST. In contrast, the large diversity of dynamic terms is not affected by global warming but by regional SST warming patterns and corresponding expansion and northward shift of the western North Pacific subtropical high, shaping moisture transport to the target subregion during different subseasons. Thus, further analysis is needed to identify detailed physical processes behind the diverse dynamic effects on regional inter-monthly precipitation patterns, as these factors will be critical in quantifying uncertainties in future precipitation projections."

Minor comments

1, Line 61: predictàprojection?

Corrected as projection.

2, Line 120, it is suggested to clarify the meaning of the subscript "REF" for better understanding.

"REF" refers to the climatology period, so we have changed it to CLIM.

3, Authors have compared the results based on CMIP5 and CMIP6. However, it seems less differences between them.

They look similar in terms of future projections but there are some noticeable differences. We have highlighted their differences in the revised manuscript throughout the section 3.1 to 3.3 (**lines 204-211, 218-222, 243-246, 255-256**).

4, Fig.1 ,Fig.2 , It should clarify how does red and blue boxes define.

The red and blue boxed indicate the timing and latitude of northward migration and peak time, respectively. We have added an explanation in the figure captions.

5, Fig.3, What does the dashed lines mean?

The dashed line indicates a value of zero percentage of relative precipitation change to climatology (1995-2014 mean). We have added an explanation in the figure caption.

**Reviewer 2**:

The manuscript shows precipitation patterns in East Asia and three subregions using matrices. It compared projections in precipitation changes between CMIP5 and CMIP6 models of different SSP scenarios. The results are important, but the manuscript does not provide a physical basis for the conclusions. I suggest including the points below before publication.

Thank you for your constructive comments. We have added more explanations and discussions responding to your points in the revised manuscript. Please find details of our responses below. Line numbers are based on the cleaned version of revised manuscript.

The basis for choosing precipitation indices (1) the time of Northward migration of the rainband and (2) the peak of the monsoon band should be explained in detail although adopted from previous literature. What other monsoon indices were reported in previous studies? What is the advantage of choosing the above two monsoon indices should be given.

Thank you for the insightful point. We have provided more detailed explanations on the precipitation indices in comparison with other East Asian monsoon circulation indices. We have also highlighted the advantage of our indices in terms of representing the inter-monthly evolution of monsoon rain bands for East Asia and its three subregions by conducting further observational analyses as follows (**lines 100-125**).

"The developed indices were first evaluated in relation to East Asia precipitation patterns (Fig. S2a,b). Figure S1 shows the regression pattern of the northward migration and peak time index over East Asia and precipitation using GPCP data from 1995 to 2014. The regression patterns reveal the movement of the monsoon precipitation band during the northward migration and peak time, indicating that these indices are suitable for representing the inter-monthly evolution of monsoon rain band in East Asia.

For further evaluations of the indices, we examined the relationships between two precipitation-based indices and two East Asia summer monsoon indices: East Asia summer monsoon index [EASMI; defined as the difference between the 850hPa zonal wind anomalies averaged over the southern (100-150°E, 10-20°N) and northern (100-150°E, 25-35°N) regions; Zhang et al., 2003] and western North Pacific subtropical high [WNPSH; defined as the 850hPa eddy geopotential height averaged over 120°-150°E, 15-30°N; Zhou et al., 2020]. The EASMI shows a statistically significant negative correlation with the peak time index over East Asia (r=-0.49 for GPCP, r=-0.46 for GMAP) and with the northward migration index over China (r=-0.45 for GPCP). In contrast, the WNPSH exhibits a strong positive correlation with the northward migration index over China (r=0.63 for GPCP, r=0.50 for CMAP). However, their correlations with indices for Korea and Japan are generally weak and not statistically significant, suggesting that these circulation-based indices have limited ability to capture regional monsoon characteristics.

This is because these summer monsoon indices are based on the large-scale atmospheric circulation during the East Asia summer, and therefore have limitations in explaining regional rainfall mechanisms and intra-seasonal variability. Figure S2c and d show the regression patterns of the northward migration and peak time indices over East Asia with the 850hPa zonal wind, which is used to calculate the EASMI. During the peak time over East Asia, zonal wind anomalies in the two regions exhibit a strong correlation with the index, whereas no significant correlation is observed during the northward migration. Figure S2e and f show the regression patterns of the northward migration and peak time indices over China with the 850hPa eddy geopotential height, which is used to calculate the WNPSH. The 850hPa eddy

geopotential height shows a strong correlation during the northward migration over China but a weaker correlation during the peak time. Overall, our proposed indices for inter-monthly precipitation evolutions have the advantage of directly reflecting precipitation changes, better representing regional features, and allowing for quantitative analysis of the intra-seasonal evolution of monsoon rain band over East Asia."

Moreover, considering your point and other reviewers' concerns, to better reflect our main findings, we have revised both the title and the terminology throughout the manuscript from "spatiotemporal precipitation pattern" to "regional inter-monthly precipitation pattern". Although the two metrics were selected based on previous studies (Kusunoki and Arakawa, 2015), we agree that the term "spatiotemporal precipitation pattern" can mislead the target of this study due to its broad meaning. We believe that "regional inter-monthly precipitation patterns" is more relevant and specific to represent what we analyze.

The scientific basis for increased precipitation in different SSP scenarios and discussions on the associated physical mechanism is shallow.

Discussions on thermodynamic effects on moisture thereby on precipitation enhancement should be elaborated.

Influence of Wind and SST changes analysis should be provided to support the results.

Thank you for the useful comment. Based on your suggestions and questions provided below, we have improved our interpretations of the physical mechanisms associated with the future precipitation change patterns. In particular, to investigate the physical processes influencing the thermodynamic and dynamic terms, we conducted an inter-model correlation analysis between future changes in these terms and changes in global mean surface temperature (GMST), low-level moisture and circulation over a long-term period, following Zhou et al (2020) and Huang et al (2022). Results indicate that GMST is found to have a strong relationship with the thermodynamic term over East Asia while the inter-model spread of the dynamic term is primarily linked to variation in low-level circulation rather than global warming. We provided detailed results and associated discussions in the new subsection 3.4 Thermodynamic and Dynamic mechanism (**lines 290-353**).

Zhou, S., Huang, G., & Huang, P. (2020). Inter-model spread of the changes in the East Asian summer monsoon system in CMIP5/6 models. Journal of Geophysical Research: Atmospheres, 125, 2020JD033016.

Huang, D., Liu, A., Zheng, Y., & Zhu, J. (2022). Inter-Model Spread of the Simulated East Asian Summer Monsoon Rainfall and the Associated Atmospheric Circulations From the CMIP6 Models. Journal of Geophysical Research: Atmospheres, 127, e2022JD037371.

How does Hadley circulation change in different SSP scenarios?

Assuming that here you indicate the local Hadley cell in western North Pacific, we have checked their changes in different SSP scenarios. Figure R1 shows the inter-model scatter plots between changes in global mean surface temperature (GMST) and changes in 500hPa omega (positive downward) for long-term (2081-2100) mean across three scenarios. The response of the Hadley circulation is not spatially uniform and we have considered two latitude zones (i.e. climatologically ascending branch of 5–20°N and descend branch of 20–30°N). During the northward migration, a significant positive correlation (r = 0.49) between GMST and ω is found over the ascending branch region, indicating that GMST increase is associated with weakened ascending (Figure R1a). However, there is no significant

correlation with GMST over the descending branch region, suggesting an unclear response to global warming (Figure R1b). Similar results are found during the peak time (Figure R1c,d). Due to the lack of a clear scenario-dependent difference in the future changes of the local Hadley circulation, particularly in its subtropical branch, it is difficult to quantify the local Hadley cell contribution to the regional inter-monthly evolution of East Asian summer monsoon. Therefore, we have not included this issue in the revised text.

[Figure]

Figure R1. Inter-model scatter plots between changes in global mean surface temperature (GMST) and changes 500hPa omega (ω, positive downward) area-averaged over (a, c) 100–150°E, 5–20°N and (b, d) 100–150°E, 20–30°N during (a, b) northward migration and (c, d) peak time based on 2081-2100 mean relative to the 1995-2014 mean. Navy, olive green, and dark red dots indicate the SSP1-2.6, SSP2-4.5, and SSP5-8.5 scenarios, respectively. Correlation coefficients across the three scenarios are provided in each panel.

Results on uncertainty factors do not quantify the uncertainty.

Thank you for your suggestion. We have quantified model uncertainty, scenario uncertainty, and internal variability in the revised text and Table S2. Additionally, to assess the magnitude of each component contributing to the overall uncertainty in the projections and to evaluate their respective contributions, we provided absolute uncertainty values (Figure S4).

Precipitation values documents in section 3.1 should be tabulated

Thank you for the good suggestion. We have added a table summarizing the future precipitation changes in the long-term period (2081-2100) during northward migration and peak time for East Asia and its subregions under SSP and RCP scenarios, and have included the table in the manuscript (Table 1).

Use of short forms M, I, TH …etc should be avoided although they are defined.

Thank you for your comment. For better readability, we have replaced all abbreviations with their full names throughout the manuscript.

**Reviewer 3:**

I recommend that the manuscript undergo major revision before publication. Below are my comments:

Thank you for your constructive comments. We have revised our manuscript responding to your points. Please find details of our responses below. Line numbers are based on the cleaned version of revised manuscript.

Based on the title and the introduction, the paper aims to provide readers with a more comprehensive and detailed understanding of the changes in the spatiotemporal distribution of East Asian summer precipitation under warming scenarios. However, the results do not achieve this goal.

The spatiotemporal precipitation pattern is a key focus of the paper, but in this study, its primary role appears to be limited to the division into two time periods. This approach leads to the following issues:

What kind of change in the spatiotemporal precipitation pattern corresponds to the different changes in these two time periods? While indices can simplify the problem, the author's conclusions should focus on the precipitation pattern. However, in the abstract, the author merely states that precipitation increases across two periods and three regions. This undermines the significance of the indices painstakingly used throughout the paper.

Thank you for the clarifying point. Considering your concern and other reviewers' questions, to better reflect our analysis and main findings, we have revised both the title and the terminology throughout the manuscript from "spatiotemporal precipitation pattern" to "regional inter-monthly precipitation pattern". Although the two metrics were selected based on previous studies (Kusunoki and Arakawa, 2015), we agree that the term "spatiotemporal precipitation pattern" can mislead the target of this study due to its broad meaning. We believe that "regional inter-monthly precipitation patterns" is more relevant and specific to represent what we analyze.

Does the authors suggest that the temporal evolution of East Asian summer monsoon precipitation can be sufficiently represented by averaging over just two time periods? If so, I think this point needs further evidence to support it.

Thank you for the good point. We agree that further evidence is needed for the representativeness of our precipitation indices for the temporal evolution of East Asian summer monsoon precipitation. In this regard, we have provided more detailed explanations on the precipitation indices in comparison with other East Asian monsoon circulation indices. We have also highlighted the advantage of our indices in terms of representing the inter-monthly evolution of monsoon rain bands for East Asia and its three subregions by conducting further observational analyses as follows (**lines 100-125**).

"The developed indices were first evaluated in relation to East Asia precipitation patterns (Fig. S2a,b). Figure S1 shows the regression pattern of the northward migration and peak time index over East Asia and precipitation using GPCP data from 1995 to 2014. The regression patterns reveal the movement of the monsoon precipitation band during the northward migration and peak time, indicating that these indices are suitable for representing the inter-monthly evolution of monsoon rain band in East Asia.

For further evaluations of the indices, we examined the relationships between two precipitation-based indices and two East Asia summer monsoon indices: East Asia summer monsoon index [EASMI; defined as the difference between the 850hPa zonal wind anomalies averaged over the southern (100-150°E, 10-20°N) and northern (100-150°E, 25-35°N) regions; Zhang et al., 2003] and western North Pacific subtropical high [WNPSH; defined as the 850hPa eddy geopotential height averaged over 120°-150°E, 15-30°N; Zhou et al., 2020]. The EASMI shows a statistically significant negative correlation with the peak time index over East Asia (r=-0.49 for GPCP, r=-0.46 for GMAP) and with the northward migration index over China (r=-0.45 for GPCP). In contrast, the WNPSH exhibits a strong positive correlation with the northward migration index over China (r=0.63 for GPCP, r=0.50 for CMAP). However, their correlations with indices for Korea and Japan are generally weak and not statistically significant, suggesting that these circulation-based indices have limited ability to capture regional monsoon characteristics.

This is because these summer monsoon indices are based on the large-scale atmospheric circulation during the East Asia summer, and therefore have limitations in explaining regional rainfall mechanisms and intra-seasonal variability. Figure S2c and d show the regression patterns of the northward migration and peak time indices over East Asia with the 850hPa zonal wind, which is used to calculate the EASMI. During the peak time over East Asia, zonal wind anomalies in the two regions exhibit a strong correlation with the index, whereas no significant correlation is observed during the northward migration. Figure S2e and f show the regression patterns of the northward migration and peak time indices over China with the 850hPa eddy geopotential height, which is used to calculate the WNPSH. The 850hPa eddy geopotential height shows a strong correlation during the northward migration over China but a weaker correlation during the peak time. Overall, our proposed indices for inter-monthly precipitation evolutions have the advantage of directly reflecting precipitation changes, better representing regional features, and allowing for quantitative analysis of the intra-seasonal evolution of monsoon rain band over East Asia."

L45-55: Regarding the uncertainties in the East Asian monsoon projections, I suggest adding the following references:

Zhou S, Huang G, Huang P. A bias-corrected projection for the changes in East Asian summer monsoon rainfall under global warming. Climate Dynamics, 2019,54(1-2): 1-16.

Zhou S, Huang G, Huang P. Changes in the East Asian summer monsoon rainfall under global warming: Moisture budget decompositions and the sources of uncertainty. Climate Dynamics, 2018, 51(4): 1363–1373.

Zhou S, Huang P, Huang G, et al. Leading source and constraint on the systematic spread of the changes in East Asian and western North Pacific summer monsoon. Environmental Research Letters, 2019, 14(12): 124059.

Thank you for informing relevant references. We have cited them in our revised manuscript to provide a more comprehensive discussion of the uncertainties in East Asian precipitation projections.